# Neural Multi-Objective Combinatorial Optimization for Flexible Job Shop Scheduling Problems

**Igor G. Smit**[1,2]**, Yaoxin Wu**[2,3,*]**Pavel Troubil**[4]**, Yingqian Zhang**[2,3] **& Wim P.M. Nuijten**[1,2]
[1]Department of Mathematics and Computer Science, Eindhoven University of Technology
[2]Eindhoven Artificial Intelligence Systems Institute, Eindhoven University of Technology
[3]Department of Industrial Engineering & Innovation Sciences, Eindhoven University of Technology
[4]Delmia R&D, Dassault Systèmes
{i.g.smit,y.wu2,yqzhang,w.p.m.nuijten}@tue.nl,pavel.troubil@3ds.com

## Abstract

Neural combinatorial optimization (NCO) has made significant advances in applying deep learning techniques to efficiently and effectively solve single-objective flexible job shop scheduling problems (FJSPs). However, the more practical multi-objective FJSPs (MOFJSPs) remain underexplored, limiting the applicability of NCO in multi-criteria decision-making scenarios. In this paper, we propose a decomposition-based NCO method to solve MOFJSPs. We present the dual conditional attention network (DCAN), a neural network architecture that takes the objective preferences along with the problem instance, aiming to learn adaptable policies over the preferences. By decomposing an MOFJSP into a set of subproblems with different preferences, the learned DCAN policies generate a set of solutions that reflect the corresponding trade-offs. We customize the Proximal Policy Optimization algorithm based on decomposition to effectively train the policy network for multiple objectives and define the state and reward based on combinations of different objectives. Extensive results showcase that our approach outperforms traditional multi-objective optimization methods and generalizes well across diverse types of problem instances.

## 1 Introduction

The flexible job shop scheduling problem (FJSP) is one of the most well-studied combinatorial optimization (CO) problems. It is a complex scheduling task where multiple jobs, each made up of ordered operations, must be processed on machines. Each operation can be performed on several alternative machines with different processing times. The goal is to create a schedule that optimizes criteria such as minimizing the makespan. FJSP has many practical applications in industries such as semiconductor manufacturing (Tamssaouet et al., 2022) and aluminum production (Zhang et al., 2016). Constraint programming (CP; Baptiste et al., 2001; Col & Teppan, 2022), heuristics (Sels et al., 2012), and metaheuristics (Rooyani & Defersha, 2019) have made great progress in solving FJSP, focusing mainly on single-objective optimization like minimizing makespan. However, real-world scenarios often involve multiple conflicting objectives, such as tardiness, flowtime, and cost.

A straightforward approach to multi-objective optimization for FJSP is to form a weighted sum of the objectives and apply single-objective methods. However, this does not provide alternative solutions leveraging trade-offs among the objectives. Thus, it is hard to choose appropriate objective weights, as the weights leading to preferred solutions vary across problem instances and scales. Hence, desired solution methods provide a Pareto set of solutions with diverse objective trade-offs. To address this issue, one can solve multiple problems with preferences using the same optimization methods. Yet, even single-objective FJSP is NP-hard, rendering such methods too computationally expensive. Instead, a more prevalent solution method is to use metaheuristics that generate a set of solutions, particularly multi-objective evolutionary algorithms. However, these metaheuristics re-

---

*Corresponding author.

quire extensive effort in manual tuning and specialized operator design to achieve good performance. Moreover, their efficiency and effectiveness tend to deteriorate as problem size increases.

Recently, neural combinatorial optimization (NCO) has attracted increasing attention to solve single-objective FJSP. NCO methods aim to learn high-quality solution policies through deep reinforcement learning (DRL), reducing reliance on heavily handcrafted strategies and enabling fast inference. NCO methods for single-objective FJSP have made great progress, mainly targeting makespan optimization (e.g., Song et al., 2022; Wang et al., 2023) and extending to variants such as dynamic or stochastic FJSPs (Zhao et al., 2024; Smit et al., 2025a).

The multi-objective FSJP (MOFJSP) has received comparatively little research exploration. Although some NCO methods are developed for simple multi-objective CO problems such as multi-objective routing problems (Lin et al., 2022; Chen et al., 2025; Li et al., 2021; Zhang et al., 2023d; Wang et al., 2024), these approaches are not applicable to the MOFJSP. They depend on episodic rewards and instance-wise gradients for policy training due to simple problem structures. However, in the context of FJSP, this leads to delayed rewards due to the long decision-making horizon, inhibiting performance. Moreover, scheduling problems have a substantially different graph structure, which requires distinct problem representations and tailored neural architectures.

We address this gap by proposing a novel decomposition-based neural multi-objective combinatorial optimization (NMOCO) method for the MOFJSP, introducing the dual conditional attention network (DCAN). DCAN employs a conditional attention mechanism that adapts operation-machine attention based on objective preferences, while relying solely on a single neural network. Furthermore, we tailor the proximal policy optimization (PPO) algorithm (Schulman et al., 2017) for multi-objective optimization by defining the state and reward functions based on different combinations of objectives. Experimental results demonstrate that the proposed method outperforms existing multi-objective optimization methods across diverse problem instances and objective combinations. Our main contributions are: (1) a decomposition-based PPO framework for multi-objective scheduling that is both theoretically grounded and practically applicable; (2) a conditional attention-based network architecture that achieves state-of-the-art performance on flexible job shop scheduling and related variants; (3) new bound-based reward functions and state features for multiple prevalent scheduling objectives, with broad applicability to other nonincreasing or nondecreasing objectives; and (4) extensive experiments showing that our approach consistently outperforms strong meta-heuristic and DRL baselines across a variety of objective combinations and problem instances.

## 2 RELATED WORK

In recent years, many NCO methods have been developed for various scheduling problems, and most use graph neural networks (GNNs) (Smit et al., 2025b). Zhang et al. (2020) created a constructive DRL approach for the job shop scheduling problem (JSSP), followed by others, who explored different network architectures and learning algorithms (e.g., Park et al., 2021; Lei et al., 2022; Kwon et al., 2021; Corsini et al., 2024; Pirnay & Grimm, 2024; van Remmerden et al., 2025a;b). For the FJSP, Song et al. (2022) first proposed a competitive end-to-end DRL algorithm. They used a heterogeneous graph and designed a heterogeneous GNN using different graph attention layers (GAT; Veličković et al., 2018) to encode machine and operation nodes. Since then, several network designs and learning structures have been proposed. For instance, Zhang et al. (2023a) integrated DRL and multi-agent RL using a multi-agent graph. Others adapted DRL methods to handle various dynamic FJSP variants (Zhao et al., 2024; Zhang et al., 2023c;b), the stochastic FJSP (Smit et al., 2025a), or different FJSP extensions (Zhang et al., 2024; Li et al., 2025). Wang et al. (2023) proposed the current state-of-the-art FJSP network architecture, using dual attention network (DAN) that comprises both self- and cross-attention, achieving superior performance over previous approaches.

There are a couple of preliminary works on the MOFJSP (Luo et al., 2021; 2022; Wu et al., 2023). However, they use a trivial vector-based state, restrict potential actions to a subset of priority dispatching rules, and limit applicability to a specific variant of the dynamic MOFJSP. Moreover, these works train only one policy that optimizes a specific trade-off point between objectives. More recently, (Su et al., 2024) proposed a method to learn different policies based on different preference vectors. However, this requires separate actor networks for each preference, resulting in a high computational cost. Moreover, their method is restricted to a specific MOFJSP with a fixed objective combination, lacking the flexibility to solve other MOFJSPs. In this paper, we propose an NCO

method that uses a single neural network to solve general MOFJSPs with distinct objectives and any combination of them.

Besides learning-based methods, many MOFJSP variants have been addressed in literature using multi-objective evolutionary algorithms, such as particle swarm optimization (Moslehi & Mahnam, 2011), the genetic algorithm NSGA-II (Deng et al., 2017; Reijnen et al., 2023), and multi-objective evolutionary algorithm based on decomposition (MOEA/D; Xiao et al., 2024). These approaches can achieve satisfactory performance for specific problem instances. However, they are highly dependent on manual tuning design and their runtimes scale poorly for larger problems.

**NMOCO for Routing**  While the MOFJSP has received little attention in NCO research, several works have recently focused on multi-objective vehicle routing problems. Li et al. (2021) proposed one of the first approaches in this area, decomposing the multi-objective problem into multiple subproblems and training a separate neural network for each. Zhang et al. (2023d) adopt a similar idea within a different meta-learning framework. However, these approaches do not scale well and do not allow adapting preference weights at inference time without retraining. Lin et al. (2022) partially address this limitation using a hypernetwork that maps objective weights to actor parameters, enabling adaptation to different preference vectors during inference, but still requiring a separate actor network per preference, limiting scalability. Subsequent works (Wang et al., 2024; Chen et al., 2025; Fan et al., 2025) move to a single-model approach, conditioning the neural model directly on the preference vector and achieving strong performance. However, these methods are tailored for routing. They build on single-objective routing architectures, use simple static coordinate-based states, and rely on REINFORCE with episodic rewards in environments that are cheap to sample. In contrast, state-of-the-art scheduling methods require stepwise rewards, richer dynamic states, and more elaborate state features, and therefore rely on actor–critic methods such as PPO. Moreover, the routing objectives are simple distance-based measures (e.g., Euclidean distances over coordinates), which differ substantially from the practically relevant objectives in scheduling. Consequently, although these works demonstrate the promise of preference-conditioned and decomposition-based NMOCO methods, their applicability to other problem classes such as scheduling is highly limited.

## 3 BACKGROUND

### 3.1 MULTI-OBJECTIVE COMBINATORIAL OPTIMIZATION

A multi-objective CO (MOCO) problem is defined as $\min_{x \in \mathcal{X}} (f_1(x), f_2(x), \ldots, f_M(x))$, where $M$ is the number of objectives and $\mathcal{X}$ is the set of feasible solutions. Since objectives are conflicting, there is no single optimal solution for all objectives. Instead, Pareto optimality is introduced.

**Definition 1** (Pareto dominance). A solution $x_1 \in \mathcal{X}$ dominates another solution $x_2 \in \mathcal{X}$ ($x_1 \prec x_2$) if and only if $f_i(x_1) \leq f_i(x_2), \forall i \in \{1, \ldots, M\}$ and $f_i(x_1) < f_i(x_2), \exists i \in \{1, \ldots, M\}$.

**Definition 2** (Pareto optimality). A solution $x^* \in \mathcal{X}$ is Pareto optimal if no other solution $x' \in \mathcal{X}$ dominates it. All Pareto optimal solutions together form the *Pareto set* $\mathcal{P} = \{x^* \in \mathcal{X} | \nexists x' \in \mathcal{X} : x' \prec x^*\}$ and their objective values form the *Pareto front* $\mathcal{F} = \{f(x) | x \in \mathcal{P}\}$.

The goal of MOCO is to find the Pareto set and its front.

**Decomposition-Based Combinatorial Optimization**  Decomposition is a popular strategy for solving MOCO problems that splits them into multiple subproblems, each being a single-objective or multi-objective problem. It provides the basis for, among others, the successful MOEA/D (Zhang & Li, 2007) method, which solves the subproblems collaboratively to construct a Pareto set. We consider the most widely used and intuitive weighted sum decomposition method (Ehrgott, 2005). Here, each subproblem minimizes a scalarized objective $min_{x \in \mathcal{X}} g(x|\boldsymbol{\lambda}) = \sum_{i=1}^{M} \lambda_i f_i(x)$, where $\boldsymbol{\lambda} \in \mathbb{R}^M$ is a preference vector such that $\lambda_i \geq 0$ and $\sum_{i=1}^{M} \lambda_i = 1$. The multi-objective problem is solved by solving $N$ subproblems that consider $N$ weight vectors.

### 3.2 MULTI-OBJECTIVE FLEXIBLE JOB SHOP SCHEDULING

The FJSP consists of a pair $(\mathcal{J}, \mathcal{M})$ where $\mathcal{J}$ is a set of jobs and $\mathcal{M}$ a set of machines. A job $J_i \in \mathcal{J}$ consists of $n_i$ operations $\mathcal{O}_i = \{O_{i1}, \ldots, O_{in_i}\}$ to be performed in order. The total set of

operations is $\mathcal{O} = \bigcup_i \mathcal{O}_i$. Each operation $O_{ij} \in \mathcal{O}$ must be processed by a single machine, selected from the set of compatible machines $\mathcal{M}_{ij} \subseteq \mathcal{M}$. The processing time of operation $O_{ij}$ on machine $M_k \in \mathcal{M}_{ij}$ is $p_{ij}^k > 0$ and each machine can process one job at a time. A solution to the FJSP is a *schedule*, which assigns a compatible machine to each operation $O_{ij}$ and determines the order of operations on each machine. While the goal of a single-objective FJSP is to find a schedule that optimizes a given objective function, the MOFJSP aims to find all schedules in the Pareto set for a given set of objectives. In this paper, we consider the makespan, total tardiness, total earliness, average flowtime, and total costs as objectives. These objectives are among the most commonly occurring in the scheduling literature (see e.g., Xie et al., 2019; Dauzère-Pérès et al., 2024) and address a variety of considerations that are relevant in practice. We define the cost of an operation to be inversely related to the processing times, so that faster operations are more expensive and vice versa. Generally, the makespan, tardiness, and flowtime objectives tend to benefit from shorter processing times, while costs and earliness can profit from some slower (but cheaper) choices. Thus, the objectives cover a range of trade-offs.

**Definition 3** (Makespan). Given the job completion times $\{C_i(x)|J_i \in \mathcal{J}\}$ in a schedule $x$, the makespan is $f_{makespan}(x) = \max_{J_i \in \mathcal{J}} C_i(x)$.

**Definition 4** (Total tardiness). Given the job completion times $\{C_i(x)|J_i \in \mathcal{J}\}$ in a schedule $x$ and the job deadlines $\{D_i|J_i \in \mathcal{J}\}$, the total tardiness is $f_{tardiness}(x) = \sum_{J_i \in \mathcal{J}} \max(C_i(x) - D_i, 0)$.

**Definition 5** (Total earliness). Given the job completion times $\{C_i(x)|J_i \in \mathcal{J}\}$ in a schedule $x$ and the job deadlines $\{D_i|J_i \in \mathcal{J}\}$, the total earliness is $f_{earliness}(x) = \sum_{J_i \in \mathcal{J}} \max(D_i - C_i(x), 0)$.

**Definition 6** (Average flowtime). Given the job completion times $\{C_i(x)|J_i \in \mathcal{J}\}$ and start times $\{S_i(x)|J_i \in \mathcal{J}\}$ in a schedule $x$, the average flowtime is $f_{flowtime}(x) = \sum_{J_i \in \mathcal{J}} (C_i(x) - S_i(x))/|\mathcal{J}|$.

**Definition 7** (Total costs). Given the processing times $\{p_{ij}(x)|O_{ij} \in \mathcal{O}\}$ in a schedule $x$ and the maximum potential processing time $p_{max}$, the total costs are $f_{costs}(x) = \sum_{O_{ij} \in \mathcal{O}} (p_{max} - p_{ij}(x))$.

## 4 METHODOLOGY

### 4.1 MARKOV DECISION PROCESS

The scheduling process involves sequential decisions, progressively assigning operations to machines. At each decision moment $t$, an operation-machine combination $(O_{ij}, M_k)$ is chosen to assign operation $O_{ij}$ to machine $M_k$. In the (multi-objective) Markov Decision Process (MDP), an agent receives the state $s_t$ that represents the partial schedule and selects an action $a_t = (O_{ij}, M_k) \in \mathcal{A}(t)$ from the feasible actions $\mathcal{A}(t)$. This set comprises the possible allocations of the first unassigned operation for each job to a compatible machine. The environment then provides reward vector $\boldsymbol{r}_t = [r_{t,1}, \ldots, r_{t,M}]$ and new state $s_{t+1}$. The schedule is completed after $|\mathcal{O}|$ actions.

**State** The relevant operations $\mathcal{O}_u(t) \subseteq \mathcal{O}$ for state $s_t$ exclude those that already have a successor scheduled on the same machine and thus do not directly influence the schedule anymore. The relevant machines $\mathcal{M}_u(t) \subseteq \mathcal{M}$ are all machines on which any of the remaining operations can be scheduled. Therefore, the state $s_t = \{\mathcal{H}_O, \mathcal{H}_M, \mathcal{H}_{OM}\}$ is defined as a triplet of operation features $\mathcal{H}_O = \{h_{O_{ij}} \in \mathbb{R}^{n_O}|O_{ij} \in \mathcal{O}_u(t)\}$, machine features $\mathcal{H}_M = \{h_{M_k} \in \mathbb{R}^{n_M}|M_k \in \mathcal{M}_u(t)\}$, and operation-machine pair features $\mathcal{H}_{OM} = \{h_{(O_{ij}, M_k)} \in \mathbb{R}^{n_{OM}}|(O_{ij}, M_k) \in \mathcal{A}(t)\}$. We refer to Wang et al. (2023) and Appendix A for a description of these features. While these features were originally proposed for makespan as a single objective, many of them are also relevant across multiple objectives. Notably, the lower bound of the completion time $\underline{C}(O_{ij}, s_t)$ is particularly relevant, as it matches directly with specific objectives. This feature allows the policy to directly monitor the measures that affect the reward, as noted in the subsequent reward formulation. Therefore, we also include the lower bound feature for each objective in the state. For total tardiness and earliness, we maintain $\underline{C}(O_{in_i}, s_t) - D_i$ for each operation. For the average flowtime, we add $\underline{F}(O_{in_i}, s_t)$ for each operation. Similarly, a cost lower bound can be included. However, due to the way we define costs, this information is already captured in existing features and there is no need to add a new feature.

**Action Space and State Transition** The action space $\mathcal{A}(t)$ consists of all compatible operation-machine pairs of the first unscheduled operations per job. By taking an action, we process an opera-

tion on a machine. The relevant operations $\mathcal{O}_u(t)$ and machines $\mathcal{M}_u(t)$ are updated and all features are updated correspondingly, giving a new state $s_{t+1}$.

**Reward** The reward $r_t = H(s_t) - H(s_{t+1})$ for an objective is inversely related to the increase in its quality measure $H(\cdot)$. Wang et al. (2023) defined the makespan quality measure using a recursively updated lower bound, which outperforms directly using the objective value because it provides a smoother signal. Concretely, they defined $\underline{C}(O_{ij}, s_t)$ which equals the scheduled completion time if the operation has been scheduled. Otherwise, it follows the recursion $\underline{C}(O_{ij}, s_t) = \underline{C}(O_{i(j-1)}, s_t) + \min_{k \in \mathcal{M}_{ij}} p_{ij}^k$. The quality measure is $H_{makespan}(s_t) = \max_{O_{ij} \in \mathcal{O}} \underline{C}(O_{ij}, s_t)$. We note that defining and maintaining a lower bound is possible for any metric that is nondecreasing during the scheduling process. For tardiness and average flowtime, we can also use the completion time lower bounds. Specifically, $H_{tardiness}(s_t) = \sum_{J_i \in \mathcal{J}} \max(\underline{C}(O_{in_i}, s_t) - D_i, 0)$ is the quality measure for total tardiness. For the average flowtime, we maintain a lower bound $\underline{F}(O_{ij}, s_t)$, which is equal to $\underline{C}(O_{ij}, s_t)$ if the first operation has not yet been scheduled. Otherwise, we have $\underline{F}(O_{ij}, s_t) = \underline{C}(O_{ij}, s_t) - S_i$. We define $H_{flowtime}(s_t) = \sum_{J_i \in \mathcal{J}} \underline{F}(O_{in_i}, s_t)/|\mathcal{J}|$. For costs, the lower bound $\underline{C}(O_{ij}, s_t)$ is the actual cost if the operation has been scheduled and the lowest possible costs between machines otherwise. Consequently, $H_{costs}(s_t) = \sum_{O_{ij} \in \mathcal{O}} \underline{C}(O_{ij}, s_t)$. Similarly, the reward can be defined for any nondecreasing objective. For earliness, we define an upper bound instead of a lowerbound, as this is a nonincreasing objective. We use $H_{earliness}(s_t) = \sum_{J_i \in \mathcal{J}} \max(D_i - \underline{C}(O_{in_i}, s_t), 0)$. Then, we can use the same $r_t$ formula to reward decreases in the upper bound. This can be done in a similar way for other nonincreasing objectives.

## 4.2 DECOMPOSITION-BASED PPO

We propose to solve the MOFJSP through a weighted sum decomposition-based PPO algorithm. We prefer weighted sum decomposition over Tchebycheff decomposition (another commonly used alternative) for two main reasons. Firstly, with weighted sum decomposition, our stepwise rewards converge to the weighted sum episodal reward (cf. Appendix B). In contrast, Tchebycheff scalarization is nonlinear and nonadditive over time, preventing this theoretical alignment. Secondly, despite having a theoretical advantage to find nonconvex fronts, Tchebycheff decomposition is empirically comparable or even inferior to the weighted sum in NCO literature (Chen et al., 2025; Wang et al., 2024). Concretely, our goal is to find a policy conditioned on the decomposed problem $\pi_\theta^*(s, \boldsymbol{\lambda})$ that maximizes its expected reward, given the problem instance and preference vector. Formally, given a distribution of problem instances $S$ and a distribution of objective preferences $\boldsymbol{\Lambda}$, we aim to find a policy $\pi_\theta^*$ such that $\pi_\theta^* = \arg\max_\pi (\mathbb{E}_{\boldsymbol{\lambda} \sim \boldsymbol{\Lambda}, s_0 \sim S}[\sum_{t=0}^{|\mathcal{O}|-1} \gamma^t \sum_{i=1}^M \lambda_i r_{t,i} | s_0, \boldsymbol{\lambda}])$. To train such policies, we propose a decomposition-based PPO algorithm (Algorithm 1). We base our method on clipped PPO with generalized advantage estimation, incorporating normalized processing times and batch normalization as suggested by (Wang et al., 2023). We generate $n_B$ problem instances for every $N_B$ episodes and for each episode we sample a new preference vector $\boldsymbol{\lambda}$ for each instance. In this way, the policy is trained using a wide variety of MOFJSP instances and multiple decomposed problems per instance. By sampling frequently and using unique preference vectors per problem instance, we prevent overfitting to specific subproblems. We ensure exploration by probabilistically sampling actions based on the output probabilities of the policy.

---

**Algorithm 1** Decomposition-based PPO

---

**Require:** Neural network with initialized parameters $\theta$
1: Sample batch of $n_B$ instances
2: **for** $n_{ep} = 1, 2, \ldots N_{ep}$ **do**
3:     **for** $b = 1, 2, \ldots n_B$ **do**                                      ▷ *In Parallel*
4:         Sample preferences $\boldsymbol{\lambda} \sim \boldsymbol{\Lambda}$
5:         **for** $t = 0, 1, \ldots, |\mathcal{O}| - 1$ **do**
6:             Sample action $a_{t,b} \sim \pi_\theta(s_{t,b}, \boldsymbol{\lambda})$
7:             Perform $a_{t,b}$ and receive $s_{t+1,b}^{det}$ and $\boldsymbol{r}_{t,b}$
8:             Compute $r_{t,b} = \boldsymbol{\lambda}^\intercal \boldsymbol{r}_{t,b}$
9:             Collect transition $(s_{t,b}, a_{t,b}, r_{t,b}, s_{t+1,b}, \boldsymbol{\lambda})$
10:         Compute generalized advantage estimates
11:     Compute PPO loss (Appendix C) and update $\theta$
12:     **if** $n_{ep}$ mod $N_B = 0$ **then** Sample $n_B$ new instances

---

## 4.3 NEURAL NETWORK ARCHITECTURE

We put forward two network architectures to learn policies $\pi_\theta$ that take the subproblem state and decomposition weights (i.e., preference vectors), based on the DAN architecture (Wang et al., 2023). We propose a straightforward yet effective technique for MOFJSP, called WI-DAN: concatenating the preference vector to the feature vectors before feeding the resulting vectors to the DAN network. Thus, we set $h_{O_{ij}} = [h_{O_{ij}}||\boldsymbol{\lambda}]$ and $h_{M_k} = [h_{M_k}||\boldsymbol{\lambda}]$. In addition, we propose the dual conditional attention network (DCAN), which uses the dual attention approach of DAN but includes a conditional attention mechanism that modifies the attention based on the objective preferences. We refer to Wang et al. (2023) for details of DAN. The proposed DCAN consists of two attention blocks: the conditional operation message attention block and the conditional machine message attention block. To facilitate understanding, we show visualizations of the network structures in Appendix D.

**Conditional Operation Message Attention Block** Each operation $O_{ij} \in \mathcal{O}_u$ has a feature vector $h_{O_{ij}}^l$ and a corresponding preference embedding $h_{\lambda_{ij}}^l$ as input to the $(l+1)$-th attention block. Especially, $h_{O_{ij}}^0$ and $h_{\lambda_{ij}}^0 = h_{\lambda_O}$ are initial linear transformations of the input features of operation $O_{ij}$ and the preference vector $\boldsymbol{\lambda}$, respectively. Using conditional attention, we update the embeddings:

$$h_{O_{ij}}^{l+1} = \sigma \left( \sum_{p=j-1}^{j+1} (\alpha_{(O_{ij},O_{ip})} \mathbf{W} h_{O_{ip}}^l) + \alpha_{(O_{ij},\lambda_{ij})} \mathbf{W} h_{\lambda_{ij}}^l \right)$$

In the equation, $\boldsymbol{\alpha}$ indicates the attention coefficients after computing the softmax over the scores $\boldsymbol{e}$, where we compute $e_{(a,b)} = \text{LeakyReLU}(\mathbf{a}^\intercal[\mathbf{W}h_a^l||\mathbf{W}h_b^l])$, $\mathbf{a}$ and $\mathbf{W}$ are learnable parameters, and $\sigma$ is a nonlinearity. Here, the attention mechanism is modified through the preference embedding. Intuitively, an artificial node, based on the preference vector, is added in the attention mechanism, alongside the operation nodes, thereby affecting the attention between the operations. These artificial node embeddings are in turn also updated in each attention block, such that they modify each block appropriately. This update is similar to the operation embedding update such that $h_{\lambda_{ij}}^{l+1} = \sigma(\sum_{p=j-1}^{j+1}(\alpha_{(\lambda_{ij},O_{ip})} \mathbf{W} h_{O_{ip}}^l) + \alpha_{(\lambda_{ij},\lambda_{ij})} \mathbf{W} h_{\lambda_{ij}}^l)$.

**Conditional Machine Message Attention Block** For each machine $M_k \in \mathcal{M}_u$ we have feature vector $h_{M_k}^l$ and we have $h_{\lambda_M}^l$, where $h_{M_k}^0$ and $h_{\lambda_M}^0$ are different linear transformations of the machine input features and the preference vector. Using conditional machine attention, we compute:

$$h_{M_k}^{l+1} = \sigma \left( \sum_{q=1}^{|\mathcal{M}_u|} (\beta_{(M_k,M_q)} \mathbf{Z} h_{M_q}^l) + \beta_{(M_k,\lambda_M)} \mathbf{Z} h_{\lambda_M}^l \right)$$

Here, $\boldsymbol{\beta}$ are the attention coefficients derived from the softmax over $\boldsymbol{u}$ with $u_{(a,b)} = \text{LeakyReLU}(\mathbf{b}^\intercal[\mathbf{Z}h_a^l||\mathbf{Z}h_b^l||\mathbf{Y}c_{(a,b)}^l])$ and $\mathbf{b}, \mathbf{Y}, \mathbf{Z}$ are learnable parameters. The coefficient $c_{(M_k,M_q)}$ is an intensity metric between machines $M_k$ and $M_q$ based on their potential operations (see Appendix E). Since such a metric does not naturally exist between a preference embedding and machine embedding, we take $c_{\lambda_M} = c_{(\cdot,\lambda_M)} = c_{(\lambda_M,\cdot)}$ as the average across all intensity metrics $c_{(M_k,M_q)}$. Analogously, the preference embedding for machine attention is also updated such that $h_{\lambda_M}^{l+1} = \sigma(\sum_{q=1}^{|\mathcal{M}_u|}(\beta_{(\lambda_M,M_q)} \mathbf{Z} h_{M_q}^l) + \beta_{(\lambda_M,\lambda_M)} \mathbf{Z} h_{\lambda_M}^l)$. The final output embeddings $h_{M_k}^L$ and $h_{O_{ij}}^L$ are used identically by the actor and critic network as in DAN.

**Critic Learning** For the critic network, instead of a single scalar output to estimate the weighted sum of objectives, we output one critic value per objective. Thus, the critic is an MLP that takes the aggregated operation and machine embeddings $h_G^L = [\frac{1}{|\mathcal{O}_u|}\sum_{O_{ij}\in\mathcal{O}_u} h_{O_{ij}}^L || \frac{1}{|\mathcal{M}_u|}\sum_{M_k\in\mathcal{M}_u} h_{M_k}^L]$ and outputs a value vector $\boldsymbol{v}(s_t) \in \mathbb{R}^M$. To train the critic, we compute the loss over all individual value estimates $\mathcal{L}_{critic} = \frac{1}{|\mathcal{O}|\cdot M}\sum_{t=0}^{|\mathcal{O}|-1}\sum_{i=1}^{M}(v_i(s_t) - \hat{r}_{t,i})^2$, where $\hat{r}_{t,i}$ are the generalized advantage estimates per reward component $r_{t,i}$. For actor training, the aggregated advantage $A_t = \sum_{i=1}^{M} \lambda_i A_{t,i}$ can still be computed before calculating the PPO loss, while it allows the critic to better attribute the losses for each objective. Although theoretically compelling, on our tests we did not find a significant performance improvement over a single-valued critic, presumably indicating that the critic value estimation is considerably less complex than the actor task.

## 5 EXPERIMENTS

**Baselines** We compare with two common multi-objective evolutionary algorithms, MOEA/D (Zhang & Li, 2007) and NSGA-II (Deb et al., 2002). Our implementation is based on the operators of Xiao et al. (2024) and Zhang et al. (2011) and the genetic algorithm from Reijnen et al. (2025). NSGA-II and MOEA/D run for 1000 generations and 80000 evaluations steps, respectively, with a population size of 100, ensuring convergence. The crossover and mutation hyperparameters follow Xiao et al. (2024) and Zhang et al. (2011). We also compare with the CP-SAT solver (Perron et al., 2023). We run CP-SAT for 1 minute per subproblem (giving a total runtime of 101 and 105 minutes per 2- and 3-objective instance, respectively). A 16-core AMD ROME 7H12 machine is used for all baselines. We also implement a hypernetwork neural network that closely follows Lin et al. (2022) and Su et al. (2024) combined with our methodology as theirs are not directly applicable.

**Problem Instances** We use the popular synthetic datasets from Song et al. (2022). Instance sizes $10\times5$, $20\times5$, $15\times10$, $20\times10$ are used for training and testing, and $30\times10$ and $40\times10$ for testing. Moreover, we evaluate on the mk (Brandimarte, 1993), rdata, edata, and vdata benchmarks (Hurink et al., 1994). For the latter three, processing times across alternative machines are the same for each operation, making costs constant and as such reducing the true number of objectives. From these datasets, the makespan, flowtime, and costs can be calculated directly. We set the deadline of job $J_i$ to $D_i = 1.5 \cdot \sum_{O_{ij} \in \mathcal{O}_i}(min_{M_k \in \mathcal{M}_{ij}} p_{ij}^k)$, similar to (Wu & Weng, 2005; Chen & Matis, 2013).

**Configurations** For training, we set $N_{ep} = 1500$, $N_B = 20$, $n_b = 20$, and evaluate once every $N_{eval} = 40$ episodes. The hyperparameters for WI-DAN and DCAN match those of DAN (Wang et al., 2023). We use 3 objective combinations: makespan-costs, tardiness-costs, and makespan-flowtime-costs. For testing, we decompose the 2-objective and 3-objective problems into 101 and 105 uniformly spread subproblems, respectively, which we solve in parallel. In training, we randomly sample preference vectors from a flat Dirichlet distribution (Ng et al., 2011). We test 100 problem instances for each synthetic dataset. For inference, we use greedy solution construction and a sampling strategy, sampling 10 solutions per subproblem. We measure performance using the normalized Hypervolume (HV; cf. Appendix F; Guerreiro et al., 2021), reporting the gap $= \frac{HV_{CP}-HV}{HV_{CP}}$ to the hypervolume of the CP-SAT approach. Moreover, we report the unique number of solutions in the found Pareto sets, the runtime, and, in Appendix G, the IGD+. We present the averages for each instance set. We use an NVIDIA A100 GPU and a 9-core Intel Xeon Platinum 8360Y CPU. Since the FJSP is a generalized scheduling problem, other problems such as the JSSP and flexible flow shop scheduling problem (FFSP) can also be solved without modifications (see Appendix H).[1]

### 5.1 RESULTS ON SYNTHETIC INSTANCES

Table 1 shows the performance of our approach for test instances matching the training sizes of the models. These results show that our method, using both WI-DAN and DCAN, learns highly competitive policies. We find that for the smallest instances, NSGA-II outperforms the DRL policies for 1 of the 3 objective combinations. For all other problem sizes and objective combinations, our approach considerably outperforms the metaheuristics, while being much faster. For $20\times10$ instances, DRL achieves a gap roughly 50% better than MOEA/D. We also observe that the gap to the CP-SAT solutions narrows for larger instances. Although both perform well and also outperform the hypernetwork approach, DCAN consistently outperforms WI-DAN. Especially for the 3-objective problems, DCAN reduces the gap by several percentage points. Moreover, DCAN consistently generates larger Pareto sets, indicating that the conditional attention mechanism improves the network's ability to exploit decomposed subproblems. Sampling further improves HV performance and Pareto set size, at the cost of higher runtime. However, the runtime is still very short compared to the baselines. In short, our DRL policies considerably outperform the NSGA-II and MOEA/D baselines, with DCAN yielding better and larger Pareto sets than the more straightforward WI-DAN. In Appendix I, we present results for a different synthetic instance set, which offers similar results. In Appendix J, we visualize multiple found Pareto sets. Although our main experiments include 2- and 3-objective problems, our methodology can handle any number of objectives. To illustrate this, we solve the 4-objective problem considering makespan, flowtime, earliness, and costs in Appendix K.

---

[1]Source code at https://github.com/ai-for-decision-making-tue/Neural_Multi-Objective_CO_for_FJSP.

Table 1: Results on synthetic instances of the same sizes as the instances used for training

| | | | Metaheuristics | | Greedy | | | Sample | | | |
| | Size | | NSGA-II | MOEA/D | Hyper | WI-DAN | DCAN | Hyper | WI-DAN | DCAN | CP-SAT |
|---|---|---|---|---|---|---|---|---|---|---|---|
| **Makespan Costs** | 10×5 | HV | 0.7208 | 0.6616 | 0.5987 | 0.6999 | 0.7104 | 0.6204 | 0.7581 | **0.7647** | 0.8313 |
| | | Gap | 13.30% | 20.42% | 27.98% | 15.81% | 14.55% | 25.37% | 8.81% | **8.01%** | 0.00% |
| | | Nr. Sol. | 8.14 | 8.81 | 6.72 | 2.91 | 3.46 | 7.78 | 7.19 | 7.77 | 13.20 |
| | | Time (s) | 247.85 | 254.85 | 0.57 | 0.57 | 0.87 | 2.26 | 2.37 | 3.00 | - |
| | 20×5 | HV | 0.5332 | 0.4807 | 0.5083 | 0.5536 | 0.5599 | 0.5268 | 0.5696 | **0.5724** | 0.6095 |
| | | Gap | 12.53% | 21.14% | 16.61% | 9.17% | 8.14% | 13.57% | 6.56% | **6.09%** | 0.00% |
| | | Nr. Sol. | 11.96 | 14.67 | 6.48 | 3.30 | 4.04 | 9.24 | 5.72 | 6.82 | 11.82 |
| | | Time (s) | 652.04 | 614.81 | 1.41 | 1.47 | 1.99 | 8.07 | 8.51 | 10.43 | - |
| | 15×10 | HV | 0.6809 | 0.5570 | 0.7039 | 0.7650 | 0.7723 | 0.7388 | 0.7920 | **0.8002** | 0.9243 |
| | | Gap | 26.33% | 39.73% | 23.85% | 17.23% | 16.44% | 20.07% | 14.31% | **13.42%** | 0.00% |
| | | Nr. Sol. | 14.12 | 16.70 | 6.95 | 8.28 | 9.14 | 9.17 | 14.16 | 15.42 | 18.16 |
| | | Time (s) | 1694.42 | 1062.99 | 2.86 | 3.07 | 4.02 | 21.19 | 22.72 | 26.98 | - |
| | 20×10 | HV | 0.6289 | 0.4879 | 0.7751 | 0.7966 | 0.8083 | 0.7236 | 0.8073 | **0.8200** | 0.8548 |
| | | Gap | 26.43% | 42.93% | 9.32% | 6.82% | 5.44% | 15.36% | 5.57% | **4.07%** | 0.00% |
| | | Nr. Sol. | 19.21 | 19.45 | 16.43 | 9.44 | 13.85 | 7.43 | 14.14 | 22.66 | 14.81 |
| | | Time (s) | 2798.44 | 1774.73 | 4.67 | 5.05 | 6.57 | 38.02 | 41.66 | 48.32 | - |
| **Tardiness Costs** | 10×5 | HV | 0.7741 | 0.7006 | 0.6335 | 0.7109 | 0.7460 | 0.7103 | 0.8102 | **0.8272** | 0.9144 |
| | | Gap | 15.35% | 23.38% | 30.72% | 22.25% | 18.42% | 22.32% | 11.40% | **9.54%** | 0.00% |
| | | Nr. Sol. | 15.41 | 9.58 | 6.52 | 2.66 | 4.36 | 11.44 | 10.28 | 12.89 | 25.81 |
| | | Time (s) | 296.60 | 256.28 | 0.59 | 0.59 | 0.89 | 2.61 | 2.66 | 3.29 | - |
| | 20×5 | HV | 0.5331 | 0.4234 | 0.5479 | 0.6372 | 0.6396 | 0.5282 | 0.6645 | **0.6700** | 0.7010 |
| | | Gap | 23.96% | 39.60% | 21.84% | 9.09% | 8.76% | 24.64% | 5.20% | **4.42%** | 0.00% |
| | | Nr. Sol. | 20.53 | 17.59 | 10.23 | 10.00 | 10.88 | 10.87 | 17.27 | 19.51 | 11.26 |
| | | Time (s) | 516.84 | 619.33 | 1.55 | 1.59 | 2.24 | 9.47 | 9.93 | 12.02 | - |
| | 15×10 | HV | 0.6745 | 0.5131 | 0.7134 | 0.7982 | 0.8094 | 0.7270 | 0.8258 | **0.8338** | 0.9505 |
| | | Gap | 29.04% | 46.02% | 24.95% | 16.03% | 14.85% | 23.51% | 13.12% | **12.28%** | 0.00% |
| | | Nr. Sol. | 22.10 | 18.96 | 13.50 | 15.45 | 17.04 | 12.10 | 26.95 | 32.39 | 20.71 |
| | | Time (s) | 1642.20 | 1048.72 | 3.12 | 3.28 | 4.45 | 23.90 | 25.69 | 30.03 | - |
| | 20×10 | HV | 0.5878 | 0.4229 | 0.7566 | 0.8100 | 0.8112 | 0.7472 | 0.8274 | **0.8306** | 0.8498 |
| | | Gap | 30.83% | 50.24% | 10.97% | 4.68% | 4.53% | 12.08% | 2.63% | **2.25%** | 0.00% |
| | | Nr. Sol. | 24.54 | 23.23 | 18.23 | 17.04 | 20.03 | 12.02 | 29.56 | 34.01 | 14.54 |
| | | Time (s) | 2869.48 | 1745.29 | 5.51 | 5.43 | 7.17 | 44.66 | 47.02 | 54.38 | - |
| **Makespan Flowtime Costs** | 10×5 | HV | **0.6146** | 0.4969 | 0.4123 | 0.4081 | 0.4647 | 0.4673 | 0.4902 | 0.5130 | 0.6831 |
| | | Gap | **10.02%** | 27.25% | 39.64% | 40.26% | 31.97% | 31.59% | 28.24% | 24.90% | 0.00% |
| | | Nr. Sol. | 167.40 | 67.01 | 23.57 | 7.33 | 22.64 | 48.43 | 39.50 | 52.84 | 64.98 |
| | | Time (s) | 229.21 | 260.88 | 0.84 | 0.82 | 1.14 | 4.99 | 4.97 | 5.66 | - |
| | 20×5 | HV | 0.3661 | 0.2595 | 0.3800 | 0.4221 | 0.4318 | 0.4021 | 0.4512 | **0.4529** | 0.5210 |
| | | Gap | 29.73% | 50.20% | 27.06% | 18.97% | 17.11% | 22.83% | 13.39% | **13.06%** | 0.00% |
| | | Nr. Sol. | 230.89 | 90.90 | 31.23 | 28.15 | 32.96 | 71.75 | 80.70 | 81.77 | 59.95 |
| | | Time (s) | 543.45 | 627.90 | 2.49 | 2.52 | 3.18 | 18.34 | 19.39 | 21.03 | - |
| | 15×10 | HV | 0.4654 | 0.3205 | 0.4844 | 0.5583 | 0.5793 | 0.4850 | 0.5782 | **0.6025** | 0.7336 |
| | | Gap | 36.57% | 56.32% | 33.97% | 23.90% | 21.03% | 33.89% | 21.18% | **17.88%** | 0.00% |
| | | Nr. Sol. | 86.70 | 37.91 | 34.10 | 29.06 | 43.96 | 75.54 | 82.73 | 121.23 | 68.32 |
| | | Time (s) | 1649.08 | 1020.61 | 4.36 | 4.52 | 5.51 | 35.56 | 37.33 | 41.00 | - |
| | 20×10 | HV | 0.3834 | 0.2547 | 0.5056 | 0.5822 | 0.6142 | 0.5025 | 0.6038 | **0.6314** | 0.6612 |
| | | Gap | 42.01% | 61.49% | 23.54% | 11.96% | 7.12% | 24.00% | 8.69% | **4.51%** | 0.00% |
| | | Nr. Sol. | 106.06 | 41.38 | 37.29 | 43.08 | 57.83 | 62.65 | 114.14 | 179.74 | 60.83 |
| | | Time (s) | 2797.74 | 1674.67 | 7.35 | 7.72 | 9.13 | 64.14 | 67.44 | 74.71 | - |

Table 2 shows the generalization to larger problem instances. We use policies trained on 20×10 instances to solve 30×10 and 40×10 instances. The results show that our policies can generalize fairly well. In fact, they outperform the baselines by an even greater margin than in Table 1. Whereas the metaheuristics and CP-SAT deteriorate quickly with larger instances, our approach retains performance. We find much better hypervolumes and larger Pareto sets for our DRL policies. WI-DAN scales better for MOFJSP with the tardiness and cost objectives, but this pattern is not consistent, as DCAN maintains its advantage for the other objective combinations. In short, our policies can transfer to larger instances, retaining the advantages over the baselines.

## 5.2 RESULTS ON BENCHMARK INSTANCES

We assess cross-distribution performance using the public benchmark datasets in Table 3. Here we present the 3-objective MOFJSP, since the 2-objective problems boil down to single-objective FJSP

Table 2: Results on large synthetic instances using the policies trained on size 20×10

| | Size | | Metaheuristics | | Greedy | | | Sample | | | CP-SAT |
|---|---|---|---|---|---|---|---|---|---|---|---|
| | | | NSGA-II | MOEA/D | Hyper | WI-DAN | DCAN | Hyper | WI-DAN | DCAN | |
| **Makespan Costs** | 30×10 | HV | 0.5477 | 0.3848 | 0.7221 | 0.7229 | 0.7437 | 0.6584 | 0.7294 | **0.7493** | 0.7425 |
| | | Gap | 26.24% | 48.18% | 2.74% | 2.64% | -0.17% | 11.33 | 1.77% | **-0.91%** | 0.00% |
| | | Nr. Sol. | 29.73 | 27.80 | 24.08 | 11.67 | 19.46 | 7.49 | 18.14 | 32.62 | 11.61 |
| | | Time (s) | 5674.45 | 3580.27 | 9.54 | 9.63 | 12.09 | 82.49 | 86.64 | 100.90 | - |
| | 40×10 | HV | 0.4832 | 0.3128 | 0.6436 | 0.6387 | 0.6629 | 0.5852 | 0.6469 | **0.6688** | 0.6326 |
| | | Gap | 23.61% | 50.54% | -1.74% | -0.97% | -4.79% | 7.48 | -2.28% | **-5.73%** | 0.00% |
| | | Nr. Sol. | 34.17 | 31.28 | 28.80 | 12.57 | 22.31 | 7.61 | 22.91 | 39.86 | 11.23 |
| | | Time (s) | 10013.26 | 6233.48 | 15.69 | 16.26 | 20.48 | 148.92 | 156.98 | 181.16 | - |
| **Tardiness Costs** | 30×10 | HV | 0.4652 | 0.3082 | 0.6676 | 0.7627 | 0.7464 | 0.6562 | **0.7694** | 0.7601 | 0.6872 |
| | | Gap | 32.31% | 55.16% | 2.85% | -10.98% | -8.60% | 4.51 | **-11.95%** | -10.61% | 0.00 |
| | | Nr. Sol. | 33.40 | 28.54 | 19.07 | 25.57 | 24.53 | 15.53 | 41.03 | 41.43 | 10.18 |
| | | Time (s) | 5664.65 | 3454.61 | 10.90 | 11.58 | 14.03 | 98.26 | 105.23 | 119.65 | - |
| | 40×10 | HV | 0.3751 | 0.2349 | 0.5708 | 0.7040 | 0.6649 | 0.5659 | **0.7084** | 0.6815 | 0.5390 |
| | | Gap | 30.41% | 56.41% | -5.89% | -30.61% | -23.35% | -4.99 | **-31.42%** | -26.43% | 0.00% |
| | | Nr. Sol. | 40.00 | 28.81 | 15.41 | 29.35 | 25.25 | 14.97 | 49.78 | 41.45 | 7.71 |
| | | Time (s) | 10020.31 | 5826.27 | 19.93 | 19.68 | 24.12 | 188.55 | 194.54 | 220.47 | - |
| **Makespan Flowtime Costs** | 30×10 | HV | 0.2789 | 0.1709 | 0.4640 | 0.5401 | 0.5707 | 0.4382 | 0.5563 | **0.5811** | 0.4766 |
| | | Gap | 41.49% | 64.13% | 2.64% | -13.33% | -19.76% | 8.06 | -16.74% | **-21.93%** | 0.00 |
| | | Nr. Sol. | 137.67 | 45.18 | 44.62 | 46.62 | 60.85 | 50.15 | 131.78 | 205.99 | 46.04 |
| | | Time (s) | 5820.54 | 3306.71 | 18.42 | 16.08 | 18.58 | 180.95 | 150.65 | 165.46 | - |
| | 40×10 | HV | 0.2022 | 0.1165 | 0.3993 | 0.4773 | 0.5087 | 0.3724 | 0.4918 | **0.5194** | 0.3043 |
| | | Gap | 33.56% | 61.73% | -31.22% | -56.84% | -67.17% | -22.39% | -61.63% | **-70.70%** | 0.00% |
| | | Nr. Sol. | 66.59 | 44.26 | 49.97 | 51.09 | 63.53 | 65.19 | 151.06 | 224.75 | 35.81 |
| | | Time (s) | 10028.12 | 5705.95 | 26.53 | 26.80 | 30.87 | 263.06 | 275.59 | 298.31 | - |

Table 3: Results on public dataset instances for the 3-objective problem using the 15×10 policies

| Size | | Metaheuristics | | Greedy | | | Sample | | | CP-SAT |
|---|---|---|---|---|---|---|---|---|---|---|
| | | NSGA-II | MOEA/D | Hyper | WI-DAN | DCAN | Hyper | WI-DAN | DCAN | |
| mk | HV | **0.3416** | 0.2517 | 0.2977 | 0.2519 | 0.2743 | 0.3172 | 0.2878 | 0.3047 | 0.5085 |
| | Gap | **32.82%** | 50.49% | 41.45% | 50.46% | 46.06% | 37.62% | 43.39% | 40.07% | 0.00% |
| | Nr. Sol. | 275.70 | 94.30 | 34.30 | 18.10 | 27.50 | 88.40 | 51.10 | 84.50 | 68.90 |
| | Time (s) | 1787.04 | 1122.09 | 4.59 | 4.78 | 5.73 | 39.05 | 41.10 | 44.05 | - |
| rdata | HV | **0.6586** | 0.5652 | 0.6004 | 0.5900 | 0.6060 | 0.6166 | 0.6033 | 0.6367 | 0.7296 |
| | Gap | **9.73%** | 22.53% | 17.71% | 19.14% | 16.94% | 15.50% | 17.31% | 12.73% | 0.00% |
| | Nr. Sol. | 8.98 | 6.98 | 5.28 | 6.08 | 5.03 | 9.00 | 9.28 | 9.48 | 15.18 |
| | Time (s) | 2120.14 | 1164.22 | 5.34 | 5.68 | 6.10 | 47.79 | 44.82 | 47.79 | - |
| edata | HV | **0.6439** | 0.5773 | 0.5256 | 0.5212 | 0.5272 | 0.5441 | 0.5427 | 0.5689 | 0.7137 |
| | Gap | **9.79%** | 19.11% | 26.36% | 26.98% | 26.14% | 23.76% | 23.96% | 20.30% | 0.00% |
| | Nr. Sol. | 10.40 | 6.70 | 4.23 | 3.95 | 3.40 | 6.23 | 6.23 | 7.43 | 16.00 |
| | Time (s) | 2106.18 | 1140.89 | 5.32 | 5.66 | 6.20 | 47.25 | 45.20 | 47.78 | - |
| vdata | HV | 0.7180 | 0.6133 | 0.6957 | 0.6799 | 0.7143 | 0.7056 | 0.6800 | **0.7378** | 0.7907 |
| | Gap | 9.20% | 22.44% | 12.02% | 14.01% | 9.66% | 10.77% | 14.00% | **6.69%** | 0.00% |
| | Nr. Sol. | 11.70 | 6.50 | 6.70 | 7.28 | 6.90 | 9.18 | 10.43 | 10.90 | 12.03 |
| | Time (s) | 2236.94 | 1189.89 | 5.34 | 5.65 | 6.20 | 47.05 | 45.85 | 48.26 | - |

for three of the four datasets (results in Appendix L). Table 3 shows NSGA-II outperforms our policies on three out of four benchmark instances, which has several reasons. Firstly, the benchmark datasets contain relatively small instances in which the metaheuristics do not yet deteriorate. Secondly, makespan and flowtime are naturally less conflicting with each other than costs. When costs are constant (in rdata, edata, and vdata), a smaller solution space contains many good solutions. Hence, metaheuristics can more easily find neighboring good solutions via genetic operations. This alleviates the weakness of these algorithms in exploring divergent search spaces. Thirdly, the metaheuristic runtime is much higher as we run them for many generations. Appendices M and N show the results with more comparable runtimes. In those scenarios with shorter runtimes, DCAN outperforms the metaheuristics, underlining its value in scenarios requiring less runtime. In addition, Appendix G shows that the DRL methods mostly outperform NSGA-II with respect to the IGD+ metric, indicating their competitiveness. Overall, DCAN and WI-DAN remain competitive, outperforming MOEA/D and achieving a good runtime-performance trade-off compared to NSGA-II.

Table 4: Results of DCAN for varying inference subproblem quantities for the 3-objective problem

| Size | | $N = 10$ Greedy | Sample | $N = 45$ Greedy | Sample | $N = 105$ Greedy | Sample | $N = 496$ Greedy | $N = 1035$ Greedy |
|---|---|---|---|---|---|---|---|---|---|
| $10 \times 5$ | HV | 0.4254 | 0.4801 | 0.4565 | 0.5033 | 0.4647 | 0.5130 | 0.4734 | 0.4771 |
| | Gap | 37.72% | 29.72% | 33.18% | 26.32% | 31.97% | 24.90% | 30.69% | 30.15% |
| | Nr. Sol. | 7.12 | 24.35 | 16.44 | 41.25 | 22.64 | 52.84 | 29.48 | 31.87 |
| | Time (s) | 0.74 | 1.16 | 0.90 | 2.70 | 1.14 | 5.66 | 2.93 | 5.57 |
| $20 \times 10$ | HV | 0.5792 | 0.6078 | 0.6056 | 0.6248 | 0.6142 | 0.6314 | 0.6269 | 0.6311 |
| | Gap | 12.41% | 8.08% | 8.41% | 5.51% | 7.12% | 4.51% | 5.19% | 4.55% |
| | Nr. Sol. | 9.37 | 51.01 | 32.40 | 123.54 | 57.83 | 179.74 | 141.37 | 195.75 |
| | Time (s) | 3.38 | 8.64 | 5.23 | 32.11 | 9.13 | 74.71 | 36.35 | 71.32 |

Table 5: Hypervolume results of ablation study. ✓indicates our proposed approach is used

| State | Reward | Makespan Costs Greedy | Sample | Tardiness Costs Greedy | Sample | Makespan Flowtime Costs Greedy | Sample |
|---|---|---|---|---|---|---|---|
| | | 0.5068 | 0.5385 | 0.5591 | 0.5417 | 0.4165 | 0.3869 |
| ✓ | | 0.5182 | 0.5383 | 0.5734 | 0.5419 | 0.4475 | 0.3908 |
| | ✓ | 0.8007 | 0.8131 | 0.7948 | 0.8111 | 0.6036 | 0.6184 |
| ✓ | ✓ | 0.8083 | 0.8200 | 0.8112 | 0.8306 | 0.6142 | 0.6314 |

## 5.3 Effect of Subproblem Quantity

We can decompose a problem into different numbers of subproblems to balance computational complexity and performance. Table 4 shows that increasing the number of subproblems increases performance, albeit with diminishing returns. It also shows that for small instances, a sampling strategy with few subproblems outperforms a greedy strategy with higher $N$. The added exploration has a big advantage in these instances. In larger instances the difference fades. Using sampling and increasing the number of subproblems by the same factor of 10 leads to similar results. Thus, tuning the number of subproblems and samples allows for a trade-off between performance and runtime, though the advantage of generating more solutions diminishes as the number of subproblems increases.

## 5.4 Ablation Study

Table 5 shows the results of the ablation study for the state features and reward formulation. Here, we solve the 20x10 instances for the three problems from Table 1 using DCAN. We compare with a simple step-wise reward without lower bounds and leaving out the proposed lower bound features. These results highlight the value of our adjustments. Especially our reward formulation is crucial to achieve good results. This reward stabilizes the reward signal and achieves better credit assignment. The added features also improve performance, although the effect is smaller than for the rewards.

## 6 Conclusion

We present a novel NMOCO approach for the MOFJSP, where we use a decomposition-based PPO algorithm to train conditional policies. These policies take both the FJSP instance and the preference vectors of the decomposed problem to determine the actions. We propose two neural networks based on straightforward preference vector input (WI-DAN) and conditional attention (DCAN). We experimentally show that the proposed approach considerably outperforms the baseline metaheuristic approaches, especially for larger instances, with DCAN outperforming WI-DAN. Our methodology can act as a base for further development of NMOCO techniques for various scheduling variants. Moreover, although we target scheduling problems, we believe components such as decomposition-based PPO, bound-based reward functions, and the conditional attention mechanism, can also be leveraged to develop NCO methods for other CO problems, providing a promising research direction for future work. Next to generalizing to a wider variety of CO problems, future work may focus on optimizing additional and more complex objectives. In addition, advanced sampling techniques specifically targeting NMOCO could help better utilize the learned policies.

ACKNOWLEDGMENTS

The LEO (Learning and Explaining Optimization) project is co-funded by Holland High Tech | TKI HSTM via the PPS allowance scheme for public-private partnerships. This work used the Dutch national e-infrastructure with the support of the SURF Cooperative using grant no. EINF-15027.

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

## A    DETAILED DESCRIPTION OF STATE FEATURES

Each state consists of operation features, machine features, and operation-machine pair features. The state features $h_{O_{ij}}$ for all $O_{ij} \in \mathcal{O}_u$ are the following:

- Minimum processing time $p_{ij}^k$ among all machines $M_k \in \mathcal{M}_{ij}$.
- Average processing time $p_{ij}^k$ among all machines $M_k \in \mathcal{M}_{ij}$.
- Span of processing times $p_{ij}^k$ among all machines $M_k \in \mathcal{M}_{ij}$.
- Proportion of machines that $O_{ij}$ can be processed on: $|\mathcal{M}_{ij}|/|\mathcal{M}|$.
- 1 if operation $O_{ij}$ is scheduled, otherwise 0.
- Number of unscheduled operations in job $J_i$.
- Sum of average processing times of all unscheduled operations in $J_i$.
- Time between when an operation became available for scheduling, and the current scheduling time in the system. 0 if the operation is not yet available for scheduling.
- Remaining processing time $p_{ij}^k$ of operation $O_{ij}$ at the current scheduling time.  0 if the operation is unscheduled.

In addition, the operation feature vector contains the relevant lower bound features, described in Section 4.1, for the objectives that are considered.

For each machine $M_k \in \mathcal{M}_u$, we have the following machine features $h_{M_k}$:

- Minimum processing time $p_{ij}^k$ among all operations $O_{ij} : M_k \in \mathcal{M}_{ij}$.
- Average processing time $p_{ij}^k$ among all operations $O_{ij} : M_k \in \mathcal{M}_{ij}$.
- Number of unscheduled operations that machine $M_k$ can process.
- Number of candidate operations that machine $M_k$ can process.
- The moment when machine $M_k$ becomes available.
- The time for which machine $M_k$ has been idle at the current scheduling moment.
- 1 if $M_k$ is processing an operation, otherwise 0.
- The remaining processing time $p_{ij}^k$ of the current processed operation $O_{ij}$ on machine $M_k$.

For each considered operation-machine pair $(O_{ij}, M_k) \in \mathcal{A}$, we use the feature vector $h_{(O_{ij}, M_k)}$:

- Processing time $p_{ij}^k$.
- Ratio of $p_{ij}^k$ to $\max_k p_{ij}^k$.
- Ratio of $p_{ij}^k$ to the maximum processing time of candidate operations that can be processed by $M_k$.
- Ratio of $p_{ij}^k$ to the maximum processing time of unscheduled operations.
- Ratio of $p_{ij}^k$ to the maximum processing time of unscheduled operations that can be processed by $M_k$.
- Ratio of $p_{ij}^k$ to the maximum processing time of the pairs in $\mathcal{A}$.
- Ratio of $p_{ij}^k$ to the remaining workload of job $J_i$.
- Sum of waiting times of $O_{ij}$ and $M_k$.

## B    THEORETICAL ALIGNMENT OF WEIGHTED SUM REWARD

**Proposition 1.** The sum of stepwise rewards is equal to the negative of the weighted sum of the increase in quality measures $H(\cdot)$, given a discounting factor $\gamma = 1$:

$$\sum_{t=0}^{|\mathcal{O}|-1} \gamma^t \sum_{i=1}^{M} \lambda_i r_{t,i} = -\sum_{i=1}^{M} \lambda_i \left( H_i(s_{|\mathcal{O}|}) - H_i(s_0) \right)$$

*Proof.*

$$\sum_{t=0}^{|\mathcal{O}|-1} \gamma^t \sum_{i=1}^{M} \lambda_i r_{t,i} = \sum_{t=0}^{|\mathcal{O}|-1} \sum_{i=1}^{M} \lambda_i r_{t,i} \quad (\gamma = 1)$$

$$= \sum_{i=1}^{M} \sum_{t=0}^{|\mathcal{O}|-1} \lambda_i r_{t,i}$$

$$= \sum_{i=1}^{M} \lambda_i \sum_{t=0}^{|\mathcal{O}|-1} r_{t,i}$$

$$= \sum_{i=1}^{M} \lambda_i \sum_{t=0}^{|\mathcal{O}|-1} (H_i(s_t) - H_i(s_{t+1}))$$

$$= -\sum_{i=1}^{M} \lambda_i \left( H_i(s_{|\mathcal{O}|}) - H_i(s_0) \right)$$

$\square$

From Proposition 1, and given that $s_0$ is a constant given by the problem instance, it follows that aiming to maximize the expected weighted sum stepwise function aligns with minimizing the increase in the weighted sum of the quality measure. Hence, optimizing our reward definition directly corresponds to optimizing our objectives.

## C  DECOMPOSITION-BASED PPO LOSS

We use a decomposition-based actor-critic clipped PPO algorithm with generalized advantage estimation (GAE). The actor loss over a given trajectory is defined as:

$$\mathcal{L}_{\text{actor}} = \frac{1}{|\mathcal{O}|} \sum_{t=0}^{|\mathcal{O}|-1} \min\left(\rho_t(\theta)A_t, \text{clip}(\rho_t(\theta), 1-\epsilon, 1+\epsilon)A_t\right)$$

$$\rho_t(\theta) = \frac{\pi_\theta(a_t|s_t, \boldsymbol{\lambda})}{\pi_{\theta_{old}}(a_t|s_t, \boldsymbol{\lambda})}$$

Here, $A_t = \sum_{i=1}^{M} \lambda_i A_{t,i}$ is the aggregated advantage estimate computed from the advantage estimates per objective and $\rho_t(\theta)$ is the output probability ratio between the current and previous policy for action $a_t$. The per-objective generalized advantage estimates follow from:

$$\delta_{t,i} = r_{t,i} + \gamma v_i(s_{t+1}) - v_i(s_t), \quad A_{t,i} = \sum_{l=0}^{|\mathcal{O}|-t-1} (\gamma\tau)^l \delta_{t+l,i}$$

Here, $v_i(s_t)$ is the value estimate for objective $i$ of the critic network, and $\gamma$ and $\tau$ are hyperparameters controlling the bias-variance trade-off of the GAE.

As explained before, the critic network is updated using the critic loss function:

$$\mathcal{L}_{\text{critic}} = \frac{1}{|\mathcal{O}| \cdot M} \sum_{t=0}^{|\mathcal{O}|-1} \sum_{i=1}^{M} (v_i(s_t) - \hat{r}_{t,i})^2$$

Here, $\hat{r}_{t,i} = A_{t,i} + v_i(s_t)$ is the bootstrapped generalized advantage estimate target for objective $i$.

The final PPO loss consists of the actor loss, critic loss, and an entropy bonus $\mathcal{L}_{\text{entropy}} = \frac{1}{|\mathcal{O}|} \sum_{t=0}^{|\mathcal{O}|-1} \mathcal{H}[\pi_\theta(\cdot|s_t, \boldsymbol{\lambda})]$ that encourages exploration:

$$\mathcal{L}_{PPO} = -\mathcal{L}_{\text{actor}} + c_1 \cdot \mathcal{L}_{\text{critic}} - c_2 \cdot \mathcal{L}_{\text{entropy}}$$

In this equation, $c_1$ and $c_2$ are coefficients that control the weights of each loss.

# D    VISUALIZATION OF NETWORK ARCHITECTURES

To further clarify the proposed WI-DAN and DCAN network architectures, we present several visual overviews in this appendix. Figure 1 offers a high-level overview of how the preference weights are incorporated in the WI-DAN and DCAN networks. In addition, Figure 2 shows a more detailed view of the conditional operation message attention block and the conditional machine message attention block of the DCAN architecture. In this figure, for simplicity, we assume a single attention head and show a single operation triplet forward pass.

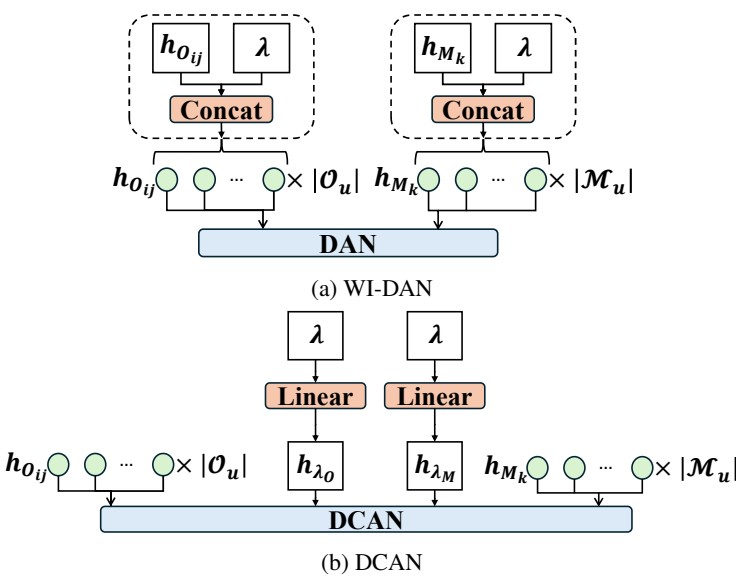

Figure 1: Visualization of high-level WI-DAN and DCAN network architectures

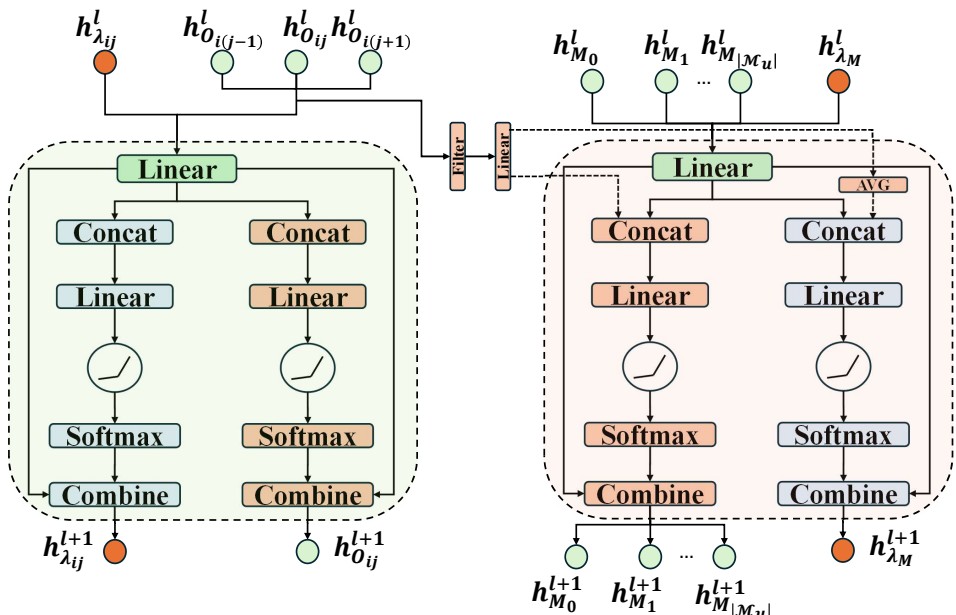

Figure 2: Detailed view of the conditional operational (left) an machine (right) message attention blocks in the DCAN network

## E  DESCRIPTION OF MACHINE INTENSITY COEFFICIENT

In the conditional machine message attention block, the coefficient $c_{(M_k, M_q)}$ is an intensity metric that measures the competition between machine $M_k$ and $M_q$. We define $\mathcal{C}_{kq}$ as the set of all operations that can be performed on both $M_k$ and $M_q$. We also define the set of candidate operations $\mathcal{J}_c = \{O_{ij} \mid \exists M_k : (O_{ij}, M_k) \in \mathcal{A}\}$ as the set of all operations that appear in at least one potential action $(O_{ij}, M_k) \in \mathcal{A}$. The intensity metric is then computed using the embeddings $h_{O_{ij}}$ of the operations in $\mathcal{C}_{kq}$:

$$c_{(M_k, M_q)} = \sum_{O_{ij} \in \mathcal{C}_{kq} \cap \mathcal{J}_c} h_{O_{ij}}$$

If $\mathcal{C}_{kq} \cap \mathcal{J}_c$ is empty, the intensity coefficient values are 0.

## F  HYPERVOLUME INDICATOR

The Hypervolume (HV) is a widely used metric for assessing performance in multi-objective optimization. Given a found Pareto front $\mathcal{F}$ and a reference point $r \in \mathbb{R}^M$, the HV is:

$$\text{HV}_r(\mathcal{F}) = \mu\left( \bigcup_{f(x) \in \mathcal{F}} [f(x), r] \right)$$

where $\mu$ denotes the Lebesgue measure, which indicates the M-dimensional volume, and $[f(x), r] = [f_1(x), r_1] \times \cdots \times [f_M(x), r_M]$ is an M-dimensional cube that spans the regions between each point $f(x)$ and the reference point $r$. The reference point is a defined point in the objective space that is typically dominated by all solutions of interest.

The HV measure is sensitive to the scales of the objectives. Hence, we report the normalized hypervolume values. To this end, we first subtract the objective lower bounds, defined by the point $z$, from the points on the Pareto front. These lower bounds are equal to the objective lower bounds in Section 4.1 at the initial state of the MDP. Then, we compute the hypervolume from these transformed points and divide by the product of the ranges between the reference point and lower bound. Thus, we use:

$$\widehat{\text{HV}}_r(\mathcal{F}) = \mu\left( \bigcup_{f(x) \in \mathcal{F}} [f(x) - z, r - z] \right) \Big/ \prod_{i=1}^{M}(r_i - z_i)$$

To define the reference point $r$ for each problem instance, we initialize 1000 solutions according to the initialization procedure of our NSGA-II approach, and take the worst value we find for each objective in this set of solutions.

## G  IGD+ PERFORMANCE METRIC

IGD+ is a performance metric for multi-objective optimization. It is defined as the average distance from each point in a given reference Pareto front to the closest point in the solution set found. The distance to the closest solution is computed using a modified Euclidean distance that only accounts for the positive part of the difference in each objective. Formally, given a reference set $\mathcal{Z}$ and a solution set $\mathcal{F}$, IGD+ is calculated as:

$$\text{IGD}^+(\mathcal{F}, \mathcal{Z}) = \frac{1}{|\mathcal{Z}|} \sum_{z \in \mathcal{Z}} \min_{f \in \mathcal{F}} d^+(z, f)$$

where

$$d^+(z, f) = \left( \sum_{i=1}^{M} \max(f_i - z_i, 0)^2 \right)^{1/2}$$

is the modified distance between the reference point $z$ and a solution point $f$. A lower IGD+ value indicates that the approximated front closely follows the reference front in both convergence and distribution.

As noted, the IGD+ requires a reference Pareto set that serves as the target. However, in our case, we do not have access to the optimal Pareto sets. Hence, to compute the IGD+ we define an alternative reference set. We construct our reference sets by taking all the solutions found by all the methods for a specific instance and using the non-dominated set among these. In this way, the resulting IGD+ measures will be influenced by this lack of a true optimal reference set, so conclusions should be drawn with care. Nevertheless, the metric still provides valuable insights alongside the hypervolume.

In Tables 6, 7, and 8, we present the IGD+ values corresponding to the main experiments. The results largely follow the same pattern observed with the hypervolume and the number of solutions in the Pareto sets. Specifically, our DRL approaches outperform the metaheuristics on the synthetic instances, with the performance gap increasing for larger instance sizes. Moreover, DCAN generally outperforms WI-DAN. However, on instances with tardiness and cost objectives, WI-DAN performs better in terms of IGD+ than in terms of hypervolume, suggesting that it produces more evenly distributed fronts in these specific cases. For the other objectives, DCAN remains superior on this metric. Another notable finding is that for the public dataset instances, our DRL policies achieve better IGD+ scores than the metaheuristics on three out of four instance sets, whereas the hypervolume was worse for all of them. This may indicate that the metaheuristics benefited from a few extreme points on the edges of the objective space, while the DRL policies generated more tightly converged solution sets. Overall, the main conclusions based on IGD+ are consistent with those drawn from the hypervolume metric, with a few noteworthy differences that provide additional insight.

## H    Results on Different Scheduling Problems

We solve JSSP and FFSP problem instances using our approach. We use the same methods and hyperparameters for these problems. For the JSSP, we train and validate on synthetic instances generated using Taillard's method (Taillard, 1993). We solve the problem with the makespan and tardiness objectives, since each operation has a fixed machine and, thus, fixed costs. Table 9 shows the results for these instances. We find that DCAN performs well for these instances. For the smallest instances, we find again that the metaheuristics are slightly better while DCAN can achieve similar performance. For larger instances, we observe that DCAN performs better again. The metaheuristics start to have trouble with the increased scale, whereas our DRL approach holds good performance while also being considerably faster. We do not outperform CP-SAT on these instances. This is sensible since the JSSP has a smaller search space than the FJSP, which means that CP-SAT will lose performance only at larger instances. We already observe that DCAN gets closer to CP-SAT for the larger sizes, while still being considerably faster.

For the FFSP, we train and validate using synthetic instances that are generated similarly to Kwon et al. (2021). We use two types of instances. One with 15 jobs and 5 stages, where the stages have 3, 2, 3, 2, and 2 machine alternatives, respectively. The other has 20 jobs and 4 stages, where each stage has 3 machine alternatives. We present these results in Table 10. Here, we see that, despite the instances being small and therefore advantageous for the metaheuristics, DCAN is competitive or advantageous over the metaheuristics in terms of hypervolume while maintaining its considerable speed advantage. It does not achieve the same hypervolume as CP-SAT. However, small instances are more suitable for CP-SAT and the runtime of DCAN is much shorter.

In short, these results confirm that our approach can be applied to other scheduling problems without modifications. We can maintain both efficiency and performance, and thereby our approach is not limited to the FJSP but can be applied to a variety of scheduling problems.

## I    Additional Synthetic Instances

Wang et al. (2023) propose an additional instance set next to the one from Song et al. (2022), which they call $SD_2$. This instance set is less realistic, as each processing time $p_{ij}^k$ is sampled uniformly from $U(1, 99)$. This implies that for the same operation, machine alternatives can be entirely different. In practice and in the synthetic data that we use, in contrast, the processing times of operations between machines are related to each other. Hence, we work with the more realistic instances in our paper. However, for completeness, we also train and test on the $SD_2$ instances using the same method. We present these results in Table 11. The results are similar to our main results, with our

DRL models outperforming the baselines considerably on most instances. In terms of hypervolume, the difference between WI-DAN and DCAN is smaller. This may be caused by the sharper decision boundaries, resulting from the unrelated processing times, that require less sophisticated differentiation between different objective preferences. However, in cases where DCAN is better, the performance improvement over WI-DAN tends to be larger than the other way around. Moreover, the DCAN generally generates a Pareto set with more unique solutions. Hence, DCAN remains beneficial over WI-DAN on these instances.

## J    VISUALIZATION OF RESULTS

To better understand the results, we visualize the Pareto fronts of one randomly selected instance per instance set of the synthetic data for the 2-objective problems. Figures 3 and 4 show these fronts. Although these figures are instance-specific and do not represent all solution shapes within each instance set, they do provide an indication of the general patterns.

The plots reflect the overall performance of the different methods, consistent with our numerical evaluation. In general, DRL policies achieve lower objective values than metaheuristics. CP-SAT solutions are highly competitive for the smaller instances, but for larger instances DRL policies tend to find better solutions. We observe that the CP-SAT solutions are generally more diverse and succeed in finding more extreme solutions at the edges of the Pareto front, strongly optimizing for one specific objective. The DRL policies, on the other hand, produce slightly more centralized solution sets. This centralization explains a significant part of the advantage CP-SAT has over DCAN and WI-DAN on smaller instances. However, for larger instances, the solutions found by the DRL are more diverse and cover a broader range of objective trade-offs.

Overall, DCAN appears to achieve a slightly wider spread of solutions than WI-DAN, which may contribute to its better hypervolume performance. All in all, our DRL approach finds well-shaped solution sets that address a meaningful range of trade-offs. Only at the extreme ends of the solution space, where one objective is heavily prioritized, does the DRL approach underperform compared to CP-SAT. However, in multi-objective optimization, trade-offs that balance the objectives are typically preferred over solutions focusing heavily on a single objective, mitigating the impact of this limitation.

## K    RESULTS ON 4-OBJECTIVE INSTANCES

We solve the 4-objective problem considering, makespan, flowtime, earliness, and costs, using 120 preferences, as presented in Table 12. These results show a similar pattern of DCAN outperforming the baselines. The gap to CP-SAT is slightly larger, which is mainly due to the fact that earliness is a non-regular objective. This is more challenging for constructive approaches, and we do not use any post-processing to allow for waiting or other adjustments in our implementation. Despite this, our approach remains superior to the metaheuristics, highlighting its ability to address problems with many objectives of differing natures.

## L    RESULTS ON BENCHMARK INSTANCES FOR 2-OBJECTIVE PROBLEMS

Table 13 shows the results on the benchmark instances for the 2-objective problems. These results show that, since the problems are reduced to single-objective problems for rdata, edata, and vdata, only one non-dominated solution is found for those. Hence, these results do not indicate multi-objective performance, but mainly which model is optimized best for makespan or tardiness. For the mk dataset, the DRL policies and NSGA-II have similar performance, with NSGA-II having slightly better hypervolume while having a much larger runtime.

## M    SHORTER INFERENCE TIMES FOR BASELINE METHODS

In the main results, we run the baseline multi-objective optimization algorithms for many generations, leading to a long runtime. For the synthetic data, our DRL approach already outperforms these algorithms with much longer runtimes. For the benchmark datasets, the NSGA-II baseline performs

slightly better. However, in practice, the available runtime is often limited. This raises the question how the performance compares when the evolutionary algorithms are given less time. Therefore, we run the NSGA-II for 50 and 100 generations, and the MOEA/D for 4000 and 8000 evaluations. Table 14 shows the results. We find that our DRL approach outperforms the baselines for similar runtimes. NSGA-II is only slightly better on the edata instances. In other instances, DCAN achieves the best performance. Thus, in these instances where our approach does not outperform the baselines when they have a longer runtime, our approach does have a better performance-runtime trade-off, making it beneficial in scheduling scenarios with limited runtimes.

## N    HIGHER NUMBER OF SAMPLES FOR DRL POLICIES

Table 15 shows the results for the benchmark instances using a higher number of samples for DCAN. We find that the performance does improve and the DRL approach becomes more competitive when given the same runtime as the NSGA-II. However, it has diminishing returns and does not provide a substantial performance boost that makes the DCAN always better than NSGA-II as can be seen in the edata instances. This is because after a certain number of samples, more duplicate solutions are produced. More elaborate search strategies can be explored to increase the test time performance of NMOCO methods.

## O    COMBINING WI-DAN AND DCAN

We also explored combining the techniques of WI-DAN with DCAN. These results are shown in Table 16. This shows that combining the methods does not lead to a clear performance increase. The conditional attention mechanism already provides a strong way to condition the policy on the objective preferences, making the additional WI mechanism redundant. Hence, for simplicity, we opted to keep them separated.

Table 6: IGD+ measures for the experiments on synthetic instances, related to Table 1

| | Size | Metaheuristics | | Greedy | | | Sample | | | |
|---|---|---|---|---|---|---|---|---|---|---|
| | | NSGA-II | MOEA/D | Hyper | WI-DAN | DCAN | Hyper | WI-DAN | DCAN | CP-SAT |
| Makespan Costs | 10×5 | 0.0713 | 0.1031 | 0.1142 | 0.0641 | 0.0574 | 0.0876 | 0.0362 | **0.0324** | 0.0030 |
| | 20×5 | 0.0489 | 0.0885 | 0.0526 | 0.0424 | 0.0367 | 0.0391 | 0.0333 | **0.0297** | 0.0063 |
| | 15×10 | 0.1434 | 0.2545 | 0.0804 | 0.0478 | 0.0451 | 0.0642 | 0.0348 | **0.0325** | 0.0058 |
| | 20×10 | 0.1414 | 0.2720 | 0.0187 | 0.0160 | 0.0094 | 0.0161 | 0.0121 | **0.0053** | 0.0137 |
| Tardiness Costs | 10×5 | 0.0833 | 0.1293 | 0.1163 | 0.0995 | 0.0748 | 0.0779 | 0.0551 | **0.0406** | 0.0001 |
| | 20×5 | 0.1237 | 0.2244 | 0.0671 | 0.0350 | 0.0277 | 0.0529 | 0.0173 | **0.0127** | 0.0111 |
| | 15×10 | 0.1796 | 0.3227 | 0.0556 | 0.0267 | 0.0219 | 0.0478 | 0.0180 | **0.0141** | 0.0202 |
| | 20×10 | 0.2049 | 0.3682 | 0.0290 | 0.0114 | 0.0137 | 0.0237 | **0.0053** | 0.0085 | 0.0452 |
| Makespan Flowtime Costs | 10×5 | **0.1044** | 0.1695 | 0.1727 | 0.1627 | 0.1390 | 0.1409 | 0.1115 | 0.1095 | 0.0005 |
| | 20×5 | 0.1560 | 0.2700 | 0.0877 | 0.0480 | 0.0473 | 0.0708 | **0.0302** | 0.0348 | 0.0109 |
| | 15×10 | 0.1506 | 0.2609 | 0.0884 | 0.0310 | 0.0253 | 0.0809 | 0.0238 | **0.0189** | 0.0052 |
| | 20×10 | 0.1757 | 0.2979 | 0.0609 | 0.0172 | 0.0172 | 0.0530 | 0.0104 | **0.0102** | 0.0225 |

Table 7: IGD+ measures for the experiments on the large synthetic instances, related to Table 2

| | Size | Metaheuristics | | Greedy | | | Sample | | | |
|---|---|---|---|---|---|---|---|---|---|---|
| | | NSGA-II | MOEA/D | Hyper | WI-DAN | DCAN | Hyper | WI-DAN | DCAN | CP-SAT |
| Makespan Costs | 30×10 | 0.1315 | 0.2959 | 0.0096 | 0.0173 | 0.0059 | 0.0081 | 0.0129 | **0.0031** | 0.0259 |
| | 40×10 | 0.1092 | 0.2716 | 0.0091 | 0.0188 | 0.0062 | 0.0076 | 0.0129 | **0.0032** | 0.0439 |
| Tardiness Costs | 30×10 | 0.2114 | 0.3786 | 0.0465 | 0.0058 | 0.0120 | 0.0430 | **0.0024** | 0.0082 | 0.0788 |
| | 40×10 | 0.1987 | 0.3573 | 0.0605 | 0.0039 | 0.0182 | 0.0575 | **0.0016** | 0.0126 | 0.1087 |
| Makespan Flowtime Costs | 30×10 | 0.1859 | 0.3134 | 0.0626 | 0.0102 | 0.0067 | 0.0796 | 0.0058 | **0.0028** | 0.0515 |
| | 40×10 | 0.1846 | 0.2971 | 0.0531 | 0.0095 | 0.0052 | 0.0510 | 0.0061 | **0.0019** | 0.0969 |

Table 8: IGD+ measures for the experiments on the public dataset instances for the 3-objective problem, related to Table 3

| Size | Metaheuristics | | Greedy | | | Sample | | | |
|---|---|---|---|---|---|---|---|---|---|
| | NSGA-II | MOEA/D | Hyper | WI-DAN | DCAN | Hyper | WI-DAN | DCAN | CP-SAT |
| mk | **0.2403** | 0.3009 | 0.2982 | 0.3213 | 0.3072 | 0.2869 | 0.2991 | 0.2917 | 0.1449 |
| rdata | 0.2182 | 0.3145 | 0.1240 | 0.1695 | 0.1160 | 0.1117 | 0.1466 | **0.0915** | 0.0310 |
| edata | 0.2108 | 0.2725 | 0.1841 | 0.2365 | 0.1835 | 0.1693 | 0.2064 | **0.1437** | 0.0304 |
| vdata | 0.2146 | 0.3067 | 0.0894 | 0.1405 | 0.0745 | 0.0812 | 0.1251 | **0.0560** | 0.0383 |

Table 9: Results on synthetic JSSP instances of the same sizes as the instances used for training for the makespan and tardiness objectives

| Size | | Metaheuristics | | Greedy | | | Sample | | | |
|---|---|---|---|---|---|---|---|---|---|---|
| | | NSGA-II | MOEA/D | Hyper | WI-DAN | DCAN | Hyper | WI-DAN | DCAN | CP-SAT |
| 6×6 | HV | 0.8703 | **0.8729** | 0.8354 | 0.7722 | 0.8002 | 0.8450 | 0.8251 | 0.8244 | 0.8778 |
| | Gap | 0.85% | **0.56%** | 4.83% | 12.02% | 8.84% | 3.73% | 6.00% | 6.07% | 0.00% |
| | IGD+ | 0.0057 | 0.0027 | 0.0291 | 0.0642 | 0.0486 | 0.0218 | 0.0322 | 0.0342 | 0.0014 |
| | Nr. Sol. | 4.04 | 4.03 | 2.73 | 1.07 | 1.56 | 3.08 | 2.24 | 2.22 | 3.17 |
| | Time (s) | 998.01 | 194.50 | 0.40 | 0.39 | 0.61 | 1.57 | 1.46 | 1.82 | - |
| 10×10 | HV | **0.8867** | 0.8592 | 0.8767 | 0.8412 | 0.8597 | 0.8844 | 0.8761 | 0.8817 | 0.9203 |
| | Gap | **3.66%** | 6.64% | 4.74% | 8.60% | 6.58% | 3.91% | 4.81% | 4.20% | 0.00% |
| | IGD+ | 0.0198 | 0.0421 | 0.0276 | 0.0473 | 0.0371 | 0.0225 | 0.0259 | 0.0242 | 0.0001 |
| | Nr. Sol. | 4.18 | 4.52 | 3.48 | 1.32 | 2.25 | 4.23 | 3.20 | 4.36 | 5.35 |
| | Time (s) | 980.67 | 662.93 | 1.66 | 1.72 | 2.35 | 10.91 | 11.69 | 13.49 | - |
| 15×15 | HV | 0.8776 | 0.7955 | 0.9003 | 0.8797 | 0.8935 | **0.9133** | 0.9074 | 0.9104 | 0.9494 |
| | Gap | 7.57% | 16.21% | 5.18% | 7.34% | 5.89% | **3.81%** | 4.42% | 4.11% | 0.00% |
| | IGD+ | 0.0459 | 0.1107 | 0.0310 | 0.0412 | 0.0347 | 0.0226 | 0.0245 | 0.0242 | 0.0000 |
| | Nr. Sol. | 4.19 | 2.85 | 3.97 | 1.37 | 2.98 | 6.34 | 4.42 | 5.92 | 13.27 |
| | Time (s) | 4823.55 | 1694.13 | 7.57 | 7.62 | 9.71 | 68.58 | 69.46 | 82.01 | - |
| 20×20 | HV | 0.8394 | 0.7791 | 0.9164 | 0.9108 | 0.9145 | 0.9239 | 0.9224 | **0.9245** | 0.9576 |
| | Gap | 12.34% | 18.65% | 4.31% | 4.89% | 4.50% | 3.52% | 3.68% | **3.46%** | 0.00% |
| | IGD+ | 0.0832 | 0.1327 | 0.0263 | 0.0291 | 0.0277 | 0.0217 | 0.0216 | 0.0215 | 0.0000 |
| | Nr. Sol. | 3.02 | 2.50 | 4.33 | 3.21 | 4.18 | 6.60 | 5.68 | 6.85 | 10.50 |
| | Time (s) | 19926.21 | 4561.18 | 28.24 | 28.29 | 30.35 | 267.84 | 275.62 | 278.54 | - |

Table 10: Results on synthetic FFSP instances of the same sizes as the instances used for training for the makespan, flowtime and costs objectives

| Size | | Metaheuristics | | Greedy | | | Sample | | | |
| --- | --- | --- | --- | --- | --- | --- | --- | --- | --- | --- |
| | | NSGA-II | MOEA/D | Hyper | WI-DAN | DCAN | Hyper | WI-DAN | DCAN | CP-SAT |
| 15×5 | HV | **0.4492** | 0.3202 | 0.3808 | 0.3898 | 0.4014 | 0.4175 | 0.4336 | 0.4385 | 0.5541 |
| | Gap | **18.92%** | 42.20% | 31.27% | 29.66% | 27.56% | 24.65% | 21.74% | 20.85% | 0.00% |
| | IGD+ | 0.0892 | 0.1878 | 0.1220 | 0.1185 | 0.1034 | 0.0986 | 0.0903 | 0.0814 | 0.0013 |
| | Nr. Sol. | 417.96 | 163.73 | 30.27 | 30.94 | 33.38 | 77.34 | 104.36 | 108.80 | 76.52 |
| | Time (s) | 639.38 | 359.20 | 1.75 | 1.73 | 2.47 | 13.26 | 13.14 | 15.05 | - |
| 20×4 | HV | 0.4039 | 0.2925 | 0.4057 | 0.3082 | 0.4200 | 0.4374 | 0.3647 | **0.4521** | 0.5424 |
| | Gap | 25.54% | 46.08% | 25.20% | 43.17% | 22.56% | 19.35% | 32.76% | **16.65%** | 0.00% |
| | IGD+ | 0.0970 | 0.1918 | 0.0553 | 0.0837 | 0.0495 | 0.0420 | 0.0557 | 0.0355 | 0.0098 |
| | Nr. Sol. | 477.05 | 207.32 | 46.69 | 13.02 | 50.03 | 151.83 | 111.49 | 177.35 | 75.71 |
| | Time (s) | 699.80 | 389.60 | 2.13 | 2.20 | 2.65 | 16.84 | 16.92 | 17.94 | - |

Table 11: Results on synthetic instances from instance set $SD_2$ of the same sizes as the instances used for training

| | Size | | Metaheuristics | | Greedy | | | Sample | | | |
| --- | --- | --- | --- | --- | --- | --- | --- | --- | --- | --- | --- |
| | | | NSGA-II | MOEA/D | Hyper | WI-DAN | DCAN | Hyper | WI-DAN | DCAN | CP-SAT |
| **Makespan Costs** | 10×5 | HV | 0.6245 | 0.5691 | 0.6130 | 0.6292 | 0.6254 | 0.6494 | **0.6740** | 0.6718 | 0.6960 |
| | | Gap | 10.27% | 18.22% | 11.93% | 9.59% | 10.14% | 6.69% | **3.16%** | 3.47% | 0.00% |
| | | IGD+ | 0.0487 | 0.0815 | 0.0452 | 0.0384 | 0.0395 | 0.0263 | 0.0167 | 0.0175 | 0.0232 |
| | | Nr. Sol. | 46.12 | 40.07 | 20.35 | 17.96 | 17.65 | 46.78 | 55.96 | 52.77 | 32.69 |
| | | Time (s) | 260.19 | 254.54 | 0.57 | 0.56 | 0.87 | 2.33 | 2.30 | 2.97 | - |
| | 20×5 | HV | 0.4554 | 0.4082 | 0.4917 | 0.3237 | 0.4926 | 0.5059 | 0.4732 | **0.5148** | 0.4875 |
| | | Gap | 6.59% | 16.28% | -0.86% | 33.60% | -1.03% | -3.77% | 2.95% | **-5.59%** | 0.00% |
| | | IGD+ | 0.0491 | 0.0858 | 0.0203 | 0.1471 | 0.0209 | 0.0122 | 0.0363 | 0.0084 | 0.0259 |
| | | Nr. Sol. | 91.79 | 61.59 | 33.04 | 9.84 | 26.67 | 91.68 | 56.30 | 83.89 | 43.82 |
| | | Time (s) | 542.25 | 617.20 | 1.45 | 1.46 | 2.13 | 7.94 | 8.46 | 10.62 | - |
| | 15×10 | HV | 0.4846 | 0.3724 | 0.5966 | 0.6312 | 0.6372 | 0.6177 | 0.6659 | **0.6677** | 0.7402 |
| | | Gap | 34.54% | 49.69% | 19.41% | 14.72% | 13.92% | 16.55% | 10.03% | **9.80%** | 0.00% |
| | | IGD+ | 0.1501 | 0.2307 | 0.0558 | 0.0413 | 0.0393 | 0.0455 | 0.0263 | 0.0258 | 0.0088 |
| | | Nr. Sol. | 96.38 | 53.51 | 25.47 | 22.27 | 22.39 | 56.81 | 64.89 | 63.71 | 54.11 |
| | | Time (s) | 1651.52 | 1033.59 | 2.92 | 2.95 | 3.94 | 21.12 | 22.93 | 26.69 | - |
| | 20×10 | HV | 0.4294 | 0.3183 | 0.6084 | 0.6332 | 0.6374 | 0.6234 | 0.6514 | **0.6540** | 0.6521 |
| | | Gap | 34.15% | 51.19% | 6.70% | 2.90% | 2.25% | 4.40% | 0.10% | **-0.29%** | 0.00% |
| | | IGD+ | 0.1458 | 0.2291 | 0.0237 | 0.0161 | 0.0149 | 0.0167 | 0.0080 | 0.0072 | 0.0167 |
| | | Nr. Sol. | 111.64 | 57.39 | 33.90 | 28.39 | 31.72 | 95.71 | 92.10 | 102.02 | 49.22 |
| | | Time (s) | 2744.02 | 1658.01 | 4.83 | 5.09 | 6.20 | 37.04 | 41.29 | 45.56 | - |
| **Tardiness Costs** | 10×5 | HV | 0.6458 | 0.5734 | 0.6210 | 0.6453 | 0.6377 | 0.6578 | **0.6893** | 0.6892 | 0.7614 |
| | | Gap | 15.18% | 24.70% | 18.44% | 15.25% | 16.25% | 13.60% | **9.48%** | 9.49% | 0.00% |
| | | IGD+ | 0.0656 | 0.1113 | 0.0681 | 0.0573 | 0.0606 | 0.0496 | 0.0345 | 0.0337 | 0.0019 |
| | | Nr. Sol. | 83.06 | 47.34 | 23.26 | 21.37 | 19.48 | 48.39 | 62.02 | 62.61 | 40.45 |
| | | Time (s) | 235.78 | 256.09 | 0.61 | 0.59 | 0.91 | 2.63 | 2.71 | 3.34 | - |
| | 20×5 | HV | 0.4736 | 0.3871 | 0.5541 | 0.5793 | 0.5842 | 0.5760 | 0.6058 | **0.6076** | 0.6226 |
| | | Gap | 23.94% | 37.83% | 11.00% | 6.95% | 6.16% | 7.48% | 2.70% | **2.40%** | 0.00% |
| | | IGD+ | 0.1039 | 0.1718 | 0.0405 | 0.0275 | 0.0250 | 0.0271 | 0.0125 | 0.0121 | 0.0086 |
| | | Nr. Sol. | 100.59 | 64.46 | 29.20 | 29.13 | 28.64 | 59.60 | 67.20 | 69.97 | 34.56 |
| | | Time (s) | 531.53 | 622.17 | 1.58 | 1.59 | 2.24 | 9.61 | 10.15 | 11.95 | - |
| | 15×10 | HV | 0.4595 | 0.3490 | 0.5922 | 0.6238 | 0.6171 | 0.6126 | **0.6515** | 0.6408 | 0.6990 |
| | | Gap | 34.26% | 50.06% | 15.27% | 10.76% | 11.72% | 12.36% | **6.80%** | 8.32% | 0.00% |
| | | IGD+ | 0.1458 | 0.2252 | 0.0381 | 0.0271 | 0.0307 | 0.0286 | 0.0163 | 0.0212 | 0.0154 |
| | | Nr. Sol. | 120.96 | 62.03 | 33.90 | 33.76 | 36.77 | 80.95 | 102.46 | 98.34 | 44.81 |
| | | Time (s) | 1651.95 | 1033.17 | 3.17 | 3.46 | 4.46 | 24.37 | 26.05 | 30.11 | - |
| | 20×10 | HV | 0.3994 | 0.2909 | 0.5916 | 0.6314 | 0.6320 | 0.6083 | **0.6507** | 0.6482 | 0.6337 |
| | | Gap | 36.98% | 54.09% | 6.65% | 0.37% | 0.26% | 4.01% | **-2.68%** | -2.28% | 0.00% |
| | | IGD+ | 0.1575 | 0.2361 | 0.0282 | 0.0138 | 0.0140 | 0.0207 | 0.0059 | 0.0071 | 0.0225 |
| | | Nr. Sol. | 123.98 | 63.14 | 38.87 | 40.15 | 41.27 | 93.95 | 109.92 | 105.89 | 37.75 |
| | | Time (s) | 2739.73 | 1655.64 | 5.41 | 5.65 | 7.01 | 42.81 | 46.89 | 52.89 | - |
| **Makespan Flowtime Costs** | 10×5 | HV | **0.4531** | 0.3789 | 0.3868 | 0.3594 | 0.3918 | 0.4092 | 0.4123 | 0.4340 | 0.5158 |
| | | Gap | **12.16%** | 26.55% | 25.01% | 30.32% | 24.05% | 20.66% | 20.07% | 15.87% | 0.00% |
| | | IGD+ | 0.0467 | 0.0822 | 0.0543 | 0.0590 | 0.0507 | 0.0399 | 0.0355 | 0.0322 | 0.0118 |
| | | Nr. Sol. | 479.91 | 193.70 | 51.90 | 23.75 | 49.21 | 137.85 | 150.34 | 183.25 | 75.62 |
| | | Time (s) | 248.22 | 260.93 | 0.83 | 0.82 | 1.11 | 5.05 | 5.03 | 5.65 | - |
| | 20×5 | HV | 0.2759 | 0.2212 | 0.3416 | 0.3383 | 0.3608 | 0.3428 | 0.3701 | **0.3827** | 0.3884 |
| | | Gap | 28.96% | 43.06% | 12.04% | 12.91% | 7.10% | 11.74% | 4.70% | **1.46%** | 0.00% |
| | | IGD+ | 0.1182 | 0.1886 | 0.0303 | 0.0287 | 0.0197 | 0.0195 | 0.0160 | 0.0116 | 0.0173 |
| | | Nr. Sol. | 692.81 | 228.96 | 62.85 | 45.76 | 64.60 | 182.79 | 220.66 | 242.45 | 87.55 |
| | | Time (s) | 565.30 | 626.18 | 2.51 | 2.60 | 3.20 | 18.76 | 19.60 | 21.41 | - |
| | 15×10 | HV | 0.2831 | 0.1992 | 0.4024 | 0.4171 | 0.4174 | 0.4217 | **0.4369** | 0.4354 | 0.4705 |
| | | Gap | 39.84% | 57.67% | 14.47% | 11.37% | 11.29% | 10.37% | **7.15%** | 7.47% | 0.00% |
| | | IGD+ | 0.1217 | 0.1904 | 0.0206 | 0.0191 | 0.0197 | 0.0146 | 0.0122 | 0.0134 | 0.0140 |
| | | Nr. Sol. | 360.96 | 120.93 | 65.32 | 46.25 | 64.42 | 262.92 | 277.51 | 289.54 | 88.82 |
| | | Time (s) | 1668.57 | 1009.72 | 4.40 | 4.57 | 5.53 | 34.83 | 37.60 | 41.35 | - |
| | 20×10 | HV | 0.2333 | 0.1597 | 0.3725 | 0.3878 | 0.3917 | 0.3874 | 0.4097 | **0.4133** | 0.3841 |
| | | Gap | 39.25% | 58.43% | 3.01% | -0.96% | -1.99% | -0.86% | -6.66% | **-7.60%** | 0.00% |
| | | IGD+ | 0.1301 | 0.2008 | 0.0250 | 0.0177 | 0.0172 | 0.0196 | 0.0117 | 0.0114 | 0.0297 |
| | | Nr. Sol. | 398.97 | 119.22 | 67.14 | 61.14 | 68.45 | 275.70 | 285.84 | 305.61 | 83.85 |
| | | Time (s) | 2757.34 | 1634.10 | 7.48 | 7.70 | 9.07 | 62.10 | 66.94 | 72.69 | - |

Table 12: Results on synthetic instances for the 4 objectives makespan, flowtime, earliness, and costs

| Size | | Metaheuristics | | Greedy | | | Sample | | | |
|---|---|---|---|---|---|---|---|---|---|---|
| | | NSGA-II | MOEA/D | Hyper | WI-DAN | DCAN | Hyper | WI-DAN | DCAN | CP-SAT |
| 10×5 | HV | **0.5628** | 0.4785 | 0.3871 | 0.3618 | 0.4333 | 0.4306 | 0.4508 | 0.4773 | 0.6531 |
| | Gap | 13.82% | 26.74% | 40.73% | 44.60/% | 33.65% | 34.07% | 30.97% | 26.92% | 0.00% |
| | IGD+ | 0.0812 | 0.1106 | 0.1344 | 0.1449 | 0.1076 | 0.1085 | 0.0966 | 0.0855 | 0.0062 |
| | Nr. Sol. | 1038.92 | 147.41 | 50.71 | 7.23 | 47.49 | 178.60 | 155.77 | 183.66 | 73.54 |
| | Time (s) | 255.13 | 267.81 | 0.92 | 0.95 | 1.24 | 5.89 | 5.87 | 6.12 | - |
| 20×5 | HV | 0.3381 | 0.2561 | 0.3545 | 0.3849 | 0.3895 | 0.3785 | **0.4177** | 0.4147 | 0.5068 |
| | Gap | 33.28% | 49.47% | 30.05% | 24.05% | 23.15% | 25.32% | 17.58% | 18.17% | 0.00% |
| | IGD+ | 0.1388 | 0.2044 | 0.0808 | 0.0661 | 0.0473 | 0.0620 | 0.0385 | 0.0326 | 0.0094 |
| | Nr. Sol. | 695.60 | 111.38 | 61.37 | 51.52 | 51.95 | 209.96 | 186.91 | 165.83 | 64.14 |
| | Time (s) | 537.58 | 629.61 | 2.87 | 2.83 | 3.51 | 23.07 | 22.90 | 23.92 | - |
| 15×10 | HV | 0.4599 | 0.3283 | 0.4447 | 0.4582 | 0.4611 | 0.4653 | 0.4768 | **0.4826** | 0.7106 |
| | Gap | 35.29% | 53.81% | 37.42% | 35.52% | 35.12% | 34.53% | 32.90% | 32.09% | 0.00% |
| | IGD+ | 0.1203 | 0.2069 | 0.1083 | 0.0979 | 0.0999 | 0.0978 | 0.0899 | 0.0902 | 0.0084 |
| | Nr. Sol. | 399.58 | 55.32 | 63.34 | 53.10 | 59.52 | 250.78 | 212.46 | 213.13 | 90.31 |
| | Time (s) | 1652.62 | 1034.72 | 5.15 | 5.01 | 6.05 | 43.59 | 43.11 | 45.12 | - |
| 20×10 | HV | 0.3828 | 0.2572 | 0.5325 | 0.5239 | 0.5500 | 0.5436 | 0.5341 | **0.5660** | 0.6485 |
| | Gap | 40.97% | 60.34% | 17.89% | 19.22% | 15.19% | 16.18% | 17.65% | 12.73% | 0.00% |
| | IGD+ | 0.1521 | 0.2736 | 0.0166 | 0.0156 | 0.0163 | 0.0129 | 0.0119 | 0.0113 | 0.0085 |
| | Nr. Sol. | 185.30 | 45.69 | 66.12 | 52.29 | 55.78 | 256.21 | 168.87 | 181.97 | 79.90 |
| | Time (s) | 2758.37 | 1697.82 | 9.36 | 9.18 | 10.33 | 81.66 | 77.99 | 82.74 | - |

Table 13: Results on public dataset instances for the 2-objective problems using the 15x10 policies

| | Size | | Metaheuristics | | Greedy | | | Sample | | | |
|---|---|---|---|---|---|---|---|---|---|---|---|
| | | | NSGA-II | MOEA/D | Hyper | WI-DAN | DCAN | Hyper | WI-DAN | DCAN | CP-SAT |
| **Makespan Costs** | mk | HV | **0.5575** | 0.4428 | 0.4844 | 0.4784 | 0.4894 | 0.5254 | 0.5231 | 0.5387 | 0.6772 |
| | | Gap | **17.67%** | 34.61% | 28.47% | 29.35% | 27.73% | 22.41% | 22.76% | 20.45% | 0.00% |
| | | IGD+ | 0.1709 | 0.3112 | 0.3014 | 0.3086 | 0.3008 | 0.2797 | 0.2829 | 0.2749 | 0.1943 |
| | | Nr. Sol. | 30.33 | 26.60 | 7.20 | 6.00 | 8.10 | 12.10 | 13.70 | 14.80 | 25.00 |
| | | Time (s) | 1437.42 | 1148.88 | 3.09 | 3.12 | 4.08 | 23.81 | 24.70 | 29.03 | - |
| | rdata | HV | 0.8128 | 0.7795 | 0.8443 | 0.8479 | 0.8473 | 0.8531 | 0.8549 | **0.8553** | 0.8642 |
| | | Gap | 5.94% | 9.79% | 2.29% | 1.89% | 1.95% | 1.27% | 1.07% | **1.03%** | 0.00% |
| | | IGD+ | 0.1286 | 0.1728 | 0.0504 | 0.0469 | 0.0474 | 0.0416 | 0.0398 | 0.0395 | 0.0306 |
| | | Nr. Sol. | 1.00 | 1.00 | 1.00 | 1.00 | 1.00 | 1.00 | 1.00 | 1.00 | 1.00 |
| | | Time (s) | 1137.94 | 1153.30 | 3.52 | 3.75 | 4.85 | 28.41 | 30.28 | 35.03 | - |
| | edata | HV | 0.7967 | 0.7799 | 0.7843 | 0.7915 | 0.7960 | 0.7943 | 0.8076 | **0.8116** | 0.8329 |
| | | Gap | 4.35% | 6.36% | 5.83% | 4.96% | 4.43% | 4.63% | 3.04% | **2.55%** | 0.00% |
| | | IGD+ | 0.1090 | 0.1410 | 0.0808 | 0.0735 | 0.0691 | 0.0639 | 0.0575 | 0.0534 | 0.0322 |
| | | Nr. Sol. | 1.00 | 1.00 | 1.00 | 1.00 | 1.00 | 1.00 | 1.00 | 1.00 | 1.00 |
| | | Time (s) | 1299.37 | 1131.10 | 3.86 | 3.73 | 4.75 | 30.15 | 30.00 | 34.68 | - |
| | vdata | HV | 0.8685 | 0.8379 | 0.9093 | 0.9104 | 0.9100 | 0.9112 | **0.9119** | 0.9114 | 0.9101 |
| | | Gap | 4.57% | 7.94% | 0.09% | -0.03% | 0.01% | -0.12% | **-0.20%** | -0.13% | 0.00% |
| | | IGD+ | 0.0997 | 0.1413 | 0.0210 | 0.0199 | 0.0202 | 0.0190 | 0.0183 | 0.0189 | 0.0202 |
| | | Nr. Sol. | 1.00 | 1.00 | 1.00 | 1.00 | 1.00 | 1.00 | 1.00 | 1.00 | 1.00 |
| | | Time (s) | 1115.27 | 1212.02 | 3.58 | 3.83 | 4.93 | 28.44 | 31.22 | 35.92 | - |
| **Tardiness Costs** | mk | HV | **0.5605** | 0.4282 | 0.5313 | 0.4867 | 0.5196 | 0.5559 | 0.5370 | 0.5548 | 0.7454 |
| | | Gap | **24.81%** | 42.55% | 28.72% | 34.71% | 30.30% | 25.42% | 27.96% | 25.58% | 0.00% |
| | | IGD+ | 0.2176 | 0.3012 | 0.2713 | 0.2961 | 0.2798 | 0.2593 | 0.2671 | 0.2558 | 0.1682 |
| | | Nr. Sol. | 47.80 | 30.60 | 14.90 | 7.90 | 11.90 | 22.60 | 17.20 | 20.90 | 31.60 |
| | | Time (s) | 1758.80 | 1129.41 | 3.46 | 3.46 | 4.43 | 28.07 | 28.08 | 32.25 | - |
| | rdata | HV | 0.8335 | 0.7726 | 0.8733 | 0.8896 | 0.8700 | 0.8811 | **0.8952** | 0.8776 | 0.9109 |
| | | Gap | 8.49% | 15.18% | 4.12% | 2.34% | 4.49% | 3.27% | **1.72%** | 3.65% | 0.00% |
| | | IGD+ | 0.2420 | 0.3381 | 0.0802 | 0.0642 | 0.0838 | 0.0727 | 0.0587 | 0.0762 | 0.0432 |
| | | Nr. Sol. | 1.00 | 1.00 | 1.00 | 1.00 | 1.00 | 1.00 | 1.00 | 1.00 | 1.00 |
| | | Time (s) | 1095.41 | 1151.52 | 3.93 | 4.17 | 5.33 | 33.11 | 34.72 | 39.60 | - |
| | edata | HV | 0.8180 | 0.7798 | 0.8262 | 0.8432 | 0.8152 | 0.8409 | **0.8574** | 0.8355 | 0.8915 |
| | | Gap | 8.24% | 12.53% | 7.33% | 5.45% | 8.56% | 5.68% | **3.82%** | 6.28% | 0.00% |
| | | IGD+ | 0.2372 | 0.3032 | 0.1117 | 0.0947 | 0.1221 | 0.0973 | 0.0808 | 0.1026 | 0.0477 |
| | | Nr. Sol. | 1.00 | 1.00 | 1.00 | 1.00 | 1.00 | 1.00 | 1.00 | 1.00 | 1.00 |
| | | Time (s) | 1164.60 | 1129.98 | 3.99 | 4.20 | 5.24 | 32.76 | 34.81 | 39.03 | - |
| | vdata | HV | 0.8848 | 0.8223 | 0.9258 | 0.9346 | 0.9177 | 0.9287 | **0.9376** | 0.9214 | 0.9354 |
| | | Gap | 5.41% | 12.10% | 1.02% | 0.09% | 1.89% | 0.72% | **-0.24%** | 1.50% | 0.00% |
| | | IGD+ | 0.2021 | 0.2993 | 0.0435 | 0.0347 | 0.0516 | 0.0406 | 0.0317 | 0.0479 | 0.0340 |
| | | Nr. Sol. | 1.00 | 1.00 | 1.00 | 1.00 | 1.00 | 1.00 | 1.00 | 1.00 | 1.00 |
| | | Time (s) | 1074.69 | 1192.19 | 4.08 | 4.21 | 5.21 | 33.00 | 34.66 | 39.50 | - |

Table 14: Results on public dataset instances for the 3-objective problem using the $15 \times 10$ policies with less generations for the baseline algorithms

| Size | | Metaheuristics | | | | Greedy | | Sample | |
|---|---|---|---|---|---|---|---|---|---|
| | | NSGA-II$_{50}$ | NSGA-II$_{100}$ | MOEA/D$_{4000}$ | MOEA/D$_{8000}$ | WI-DAN | DCAN | WI-DAN | DCAN |
| mk | HV | 0.2522 | 0.2839 | 0.1962 | 0.2141 | 0.2519 | 0.2743 | 0.2878 | **0.3047** |
| | Gap | 50.40% | 44.16% | 61.41% | 57.90% | 50.46% | 46.06% | 43.39% | **40.07%** |
| | IGD+ | 0.2924 | 0.2766 | 0.3399 | 0.3323 | 0.3213 | 0.3072 | 0.2991 | 0.2917 |
| | Nr. Sol. | 85.60 | 123.90 | 49.00 | 57.80 | 18.10 | 27.50 | 51.10 | 84.50 |
| | Time (s) | 87.24 | 199.54 | 55.92 | 112.11 | 4.78 | 5.73 | 41.10 | 44.05 |
| rdata | HV | 0.5572 | 0.5904 | 0.4940 | 0.5145 | 0.5900 | 0.5809 | 0.6033 | **0.6052** |
| | Gap | 23.63% | 19.08% | 32.29% | 29.48% | 19.14% | 20.39% | 17.31% | **17.06%** |
| | IGD+ | 0.3125 | 0.2829 | 0.3703 | 0.3538 | 0.1695 | 0.1160 | 0.1466 | 0.0915 |
| | Nr. Sol. | 7.05 | 7.50 | 5.53 | 5.35 | 6.08 | 6.73 | 9.28 | 8.90 |
| | Time (s) | 103.34 | 205.98 | 57.21 | 114.50 | 5.68 | 6.86 | 50.05 | 54.29 |
| edata | HV | 0.5594 | **0.5861** | 0.5201 | 0.5297 | 0.5212 | 0.5207 | 0.5427 | 0.5463 |
| | Gap | 21.62% | **17.89%** | 27.12% | 25.78% | 26.98% | 27.04% | 23.96% | 23.46% |
| | IGD+ | 0.2856 | 0.2649 | 0.3236 | 0.3194 | 0.2365 | 0.1835 | 0.2064 | 0.1437 |
| | Nr. Sol. | 6.73 | 8.00 | 5.20 | 5.48 | 3.95 | 4.23 | 6.23 | 5.95 |
| | Time (s) | 103.29 | 207.07 | 56.16 | 112.97 | 5.66 | 6.80 | 49.50 | 54.21 |
| vdata | HV | 0.6340 | 0.6556 | 0.5436 | 0.5646 | 0.6799 | 0.6701 | 0.6800 | **0.6855** |
| | Gap | 19.82% | 17.09% | 31.25% | 28.59% | 14.01% | 15.25% | 14.00% | **13.30%** |
| | IGD+ | 0.2906 | 0.2705 | 0.3666 | 0.3531 | 0.1405 | 0.0745 | 0.1251 | 0.0560 |
| | Nr. Sol. | 7.60 | 8.78 | 5.175 | 5.85 | 7.28 | 7.98 | 10.43 | 10.85 |
| | Time (s) | 104.50 | 208.22 | 60.10 | 120.29 | 5.65 | 6.81 | 48.82 | 54.10 |

Table 15: Results on public dataset instances for the 3-objective problem using the $15 \times 10$ policies with more inference samples per preference

| Size | | Metaheuristics | | Sample | | | CP-SAT |
|---|---|---|---|---|---|---|---|
| | | NSGA-II | MOEA/D | 10 | 100 | 400 | |
| mk | HV | 0.3416 | 0.2517 | 0.3047 | 0.3258 | 0.3351 | 0.5085 |
| | Gap | 32.82% | 50.49% | 40.07% | 35.93% | 34.09% | 0.00% |
| | IGD+ | 0.2403 | 0.3009 | 0.2917 | 0.2753 | 0.2697 | 0.1449 |
| | Nr. Sol. | 275.70 | 94.30 | 84.50 | 162.30 | 200.40 | 68.90 |
| | Time (s) | 1787.04 | 1122.09 | 44.05 | 411.15 | 1647.12 | - |
| rdata | HV | 0.6586 | 0.5652 | 0.6367 | 0.6487 | 0.6561 | 0.7296 |
| | Gap | 9.73% | 22.53% | 12.73% | 11.10% | 10.07% | 0.00% |
| | IGD+ | 0.2182 | 0.3145 | 0.0915 | 0.0810 | 0.0754 | 0.0310 |
| | Nr. Sol. | 8.98 | 6.98 | 9.48 | 11.72 | 14.28 | 15.18 |
| | Time (s) | 2120.14 | 1164.22 | 47.79 | 485.12 | 1938.43 | - |
| edata | HV | 0.6439 | 0.5773 | 0.5689 | 0.5811 | 0.5883 | 0.7137 |
| | Gap | 9.79% | 19.11% | 20.30% | 18.58% | 17.58% | 0.00% |
| | IGD+ | 0.2108 | 0.2725 | 0.1437 | 0.1326 | 0.1270 | 0.0304 |
| | Nr. Sol. | 10.40 | 6.70 | 7.43 | 9.40 | 10.55 | 16.00 |
| | Time (s) | 2106.18 | 1140.89 | 47.78 | 486.01 | 1944.45 | - |
| vdata | HV | 0.7180 | 0.6133 | 0.7378 | 0.7474 | 0.7524 | 0.7907 |
| | Gap | 9.20% | 22.44% | 6.69% | 5.47% | 4.84% | 0.00% |
| | IGD+ | 0.2146 | 0.3067 | 0.0560 | 0.0482 | 0.0451 | 0.0383 |
| | Nr. Sol. | 11.70 | 6.50 | 10.90 | 12.60 | 15.15 | 12.03 |
| | Time (s) | 2236.94 | 1189.89 | 48.26 | 487.44 | 1957.78 | - |

Table 16: Performance comparison between DCAN and WI-DCAN, the architecture combining the ideas of WI-DAN and DCAN

| | | | 10x5 | | 20x5 | | 15x10 | | 20x10 | |
| --- | --- | --- | --- | --- | --- | --- | --- | --- | --- | --- |
| | | | Greedy | Sample | Greedy | Sample | Greedy | Sample | Greedy | Sample |
| Makespan Costs | DCAN | HV | 0.7104 | 0.7647 | 0.5599 | 0.5724 | 0.7723 | 0.8002 | 0.8083 | 0.8200 |
| | | Nr. Sol. | 3.46 | 7.77 | 4.04 | 6.82 | 9.14 | 15.42 | 13.85 | 22.66 |
| | WI-DCAN | HV | 0.7255 | 0.7644 | 0.5571 | 0.5716 | 0.7746 | 0.8012 | 0.8122 | 0.8209 |
| | | Nr. Sol. | 4.45 | 8.45 | 4.31 | 6.86 | 9.60 | 16.22 | 15.61 | 24.72 |
| Tardiness Costs | DCAN | HV | 0.7460 | 0.8272 | 0.6396 | 0.6700 | 0.8094 | 0.8338 | 0.8112 | 0.8306 |
| | | Nr. Sol. | 4.36 | 12.89 | 10.88 | 19.51 | 17.04 | 32.39 | 20.03 | 34.01 |
| | WI-DCAN | HV | 0.7668 | 0.8310 | 0.6359 | 0.6673 | 0.7914 | 0.8211 | 0.8147 | 0.8295 |
| | | Nr. Sol. | 6.70 | 16.22 | 8.08 | 16.48 | 14.28 | 29.48 | 20.37 | 34.71 |
| Makespan Flowtime Costs | DCAN | HV | 0.4647 | 0.5130 | 0.4318 | 0.4529 | 0.5793 | 0.6025 | 0.6142 | 0.6314 |
| | | Nr. Sol. | 22.64 | 52.84 | 32.96 | 81.77 | 43.96 | 121.23 | *57.83* | 179.74 |
| | WI-DCAN | HV | 0.4988 | 0.5484 | 0.4028 | 0.4307 | 0.5883 | 0.6061 | 0.6015 | 0.6170 |
| | | Nr. Sol | 23.6 | 65.77 | 33.52 | 79.74 | 44.56 | 112.75 | 55.03 | 175.13 |

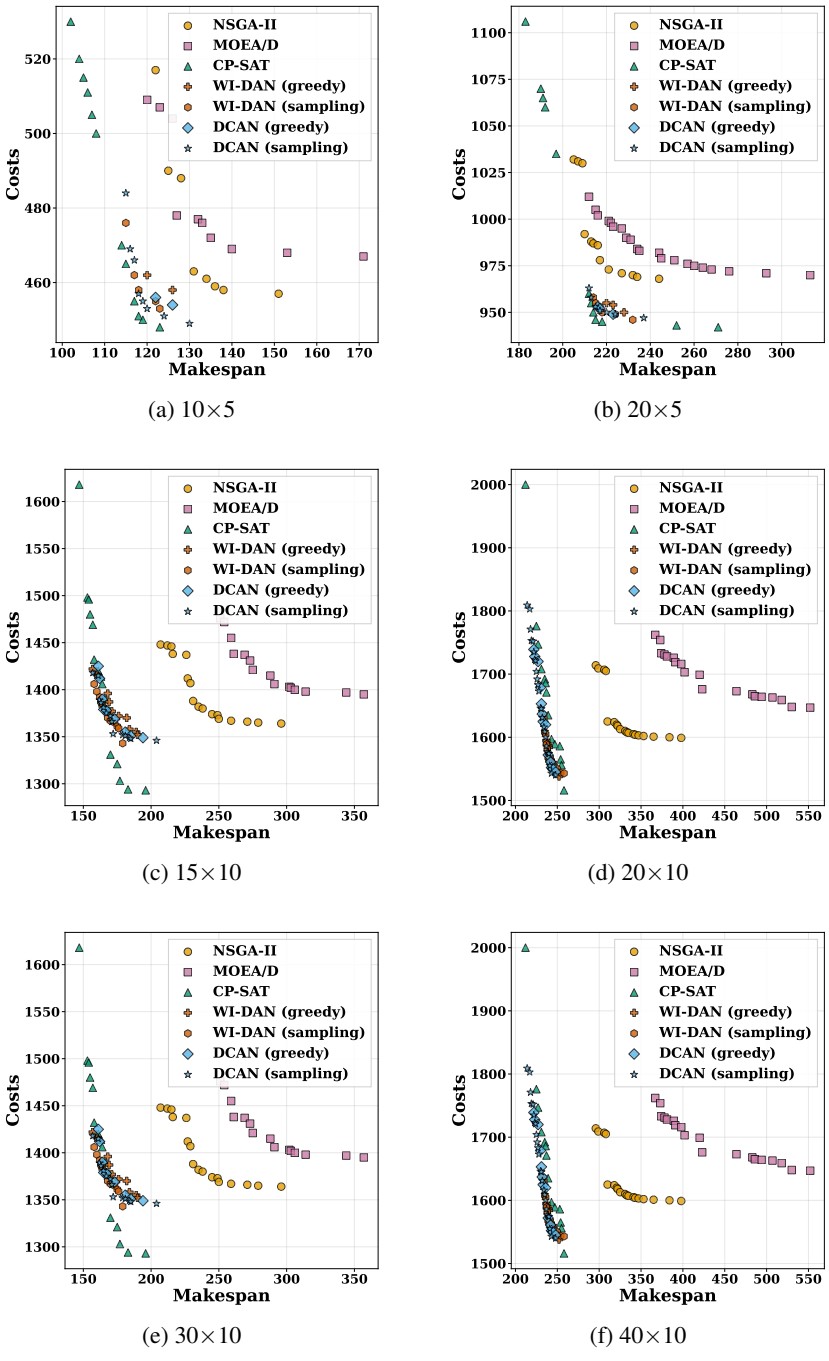

Figure 3: Visualization of solutions of randomly selected instances from different sizes for the makespan and costs

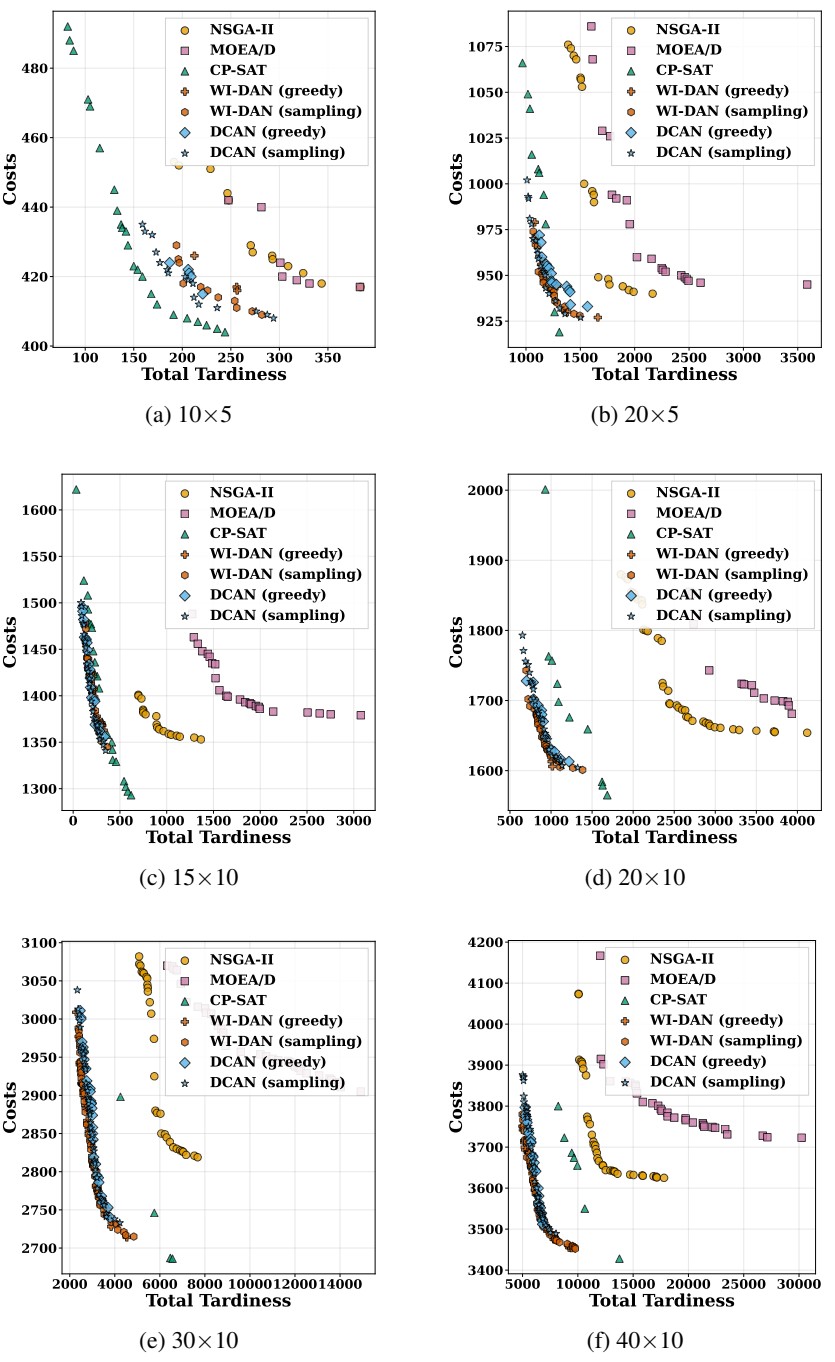

Figure 4: Visualization of solutions of randomly selected instances from different sizes for the tardiness and costs

