# OpenReview forum: "Neural Multi-Objective Combinatorial Optimization for Flexible Job Shop Scheduling Problems"
_ICLR.cc/2026/Conference — ICLR 2026 Poster_

### Official Review · Reviewer_XAsV · 2025-10-24

**Soundness:** 2
**Presentation:** 2
**Contribution:** 2
**Rating:** 4
**Confidence:** 4

**Summary:**

This paper addresses the flexible job shop scheduling problem with multiple objectives. The authors propose a Dual Conditional Attention Network that learns adaptable scheduling policies. They employ a decomposition-based neural combinatorial optimization approach, solving subproblems with different objective preferences, and use PPO for training. The learned policy can generate a set of Pareto-optimal solutions.

**Strengths:**

The paper tackles a practically important and realistic problem, as real-world scheduling problems typically involve multiple conflicting objectives.

The approach of learning adaptable policies conditioned on objective preferences is promising and could improve generalization.

Generating a Pareto set of solutions using a single trained model is an attractive feature for practical multi-objective scheduling.

**Weaknesses:**

The four considered objectives (makespan, total tardiness, average flow time, and total cost) are not well justified. The first three are closely correlated regular measures, while the inclusion of total cost lacks sufficient motivation or explanation regarding its relevance in real-world FJSPs.

The paper does not analyze the interrelationship among the four objectives, which limits understanding of the trade-offs the model learns.

While the paper presents an interesting approach, the overall contribution remains unclear. It would be helpful to explicitly highlight what new insight or technical novelty it brings beyond existing multi-objective scheduling and NCO frameworks.

The method uses the lower bound of completion time as the state and reward, but this can cause instability in learning depending on the tightness of the bound. This issue should be discussed in detail.

While several performance-enhancing techniques are proposed, the paper lacks an ablation study to quantify their individual contributions.

**Questions:**

1. How can practical constraints such as setup times or machine dedication be incorporated into your framework?
2. How does the model perform when including objectives that are not regular measures (e.g., earliness)?
3. Many recent studies employ REINFORCE for scheduling. What motivated your choice of PPO?
4. Using lower bounds as rewards can lead to inconsistent gradients depending on their tightness. How do you handle or mitigate this issue?
5. Please provide an ablation study to analyze the contribution of each proposed component.

---

> ### Author Response · Authors · 2025-11-25
> **Response to Reviewer XAsV (1/7)**
>
> **We thank the reviewer for the effort in reading our paper and for the comments. We are pleased that the reviewer acknowledges the relevance of our work and appreciates the attractiveness of our conditional network approach and experimental results. Below, we address the raised comments.**
>
> **W1**: We have included additional explanation of our choice of objectives in Section 3.2 in the revised manuscript. The objectives we consider form a representative set that is prevalent in the literature and applicable in many practical scenarios, as highlighted in surveys such as [1,2] (see, for example, Table 1 in [2]). Makespan, flowtime, and tardiness are among the most commonly studied objectives in the scheduling literature. They directly relate to practical concerns such as minimizing total completion time, reducing work-in-progress inventory, and meeting due dates, which are all crucial to logistical success. Moreover, the cost objective is motivated by studies such as [3,4] and discussions with industry experts. It is common in practice to have both expensive and cheaper production alternatives, and minimizing costs is a highly relevant objective in such scenarios. In fact, for many companies, cost minimization is a primary concern.
>
> Makespan, flowtime, and tardiness are indeed correlated measures, as we already noted in line 425 in Section 5.2 of the original manuscript. However, they still have competing interests. For example, one can minimize flowtime by scheduling all jobs as much sequentially as possible so as to not have waiting between the operations, but this will lead to higher makespans as other jobs are not efficiently processed in parallel. In addition, the cost objective competes strongly with these time-based objectives, and in each experiment we combine at least one of the former three with cost to ensure meaningful trade-offs.
>
> To further address the reviewer's concern, we have now also included an additional earliness objective, which likewise competes with objectives such as makespan and flowtime. We added a bound-based reward function for earliness in Section 4.1 and included experiments with the earliness objective in Appendix K and Table 12. The results (also given below) show that our approach continues to outperform the baselines.
>
> |              |          | Metaheuristics |         | Greedy  |          |         | Sample  |         |         |        |
> |--------------|----------|----------------|---------|---------|----------|---------|---------|---------|---------|--------|
> | Size         |          | NSGA-II        | MOEA/D  | Hyper   | WI-DAN   | DCAN    | Hyper   | WI-DAN  | DCAN    | CP-SAT |
> | 10×5  | HV       | 0.5628         | 0.4785  | 0.3871  | 0.3618   | 0.4333  | 0.4306  | 0.4508  | 0.4773  | 0.6531 |
> |              | Gap      | 13.82\%        | 26.74\% | 40.73\% | 44.60\% | 33.65\% | 34.07\% | 30.97\% | 26.92\% | 0.00\% |
> |              | Nr. Sol. | 1038.92        | 147.41  | 50.71   | 7.23     | 47.49   | 178.60  | 155.77  | 183.66  | 73.54  |
> |              | Time (s) | 255.13         | 267.81  | 0.92    | 0.95     | 1.24    | 5.89    | 5.87    | 6.12    | -      |
> | 20×5  | HV       | 0.3381         | 0.2561  | 0.3545  | 0.3849   | 0.3895  | 0.3785  | 0.4177  | 0.4147  | 0.5068 |
> |              | Gap      | 33.28\%        | 49.47\% | 30.05\% | 24.05\%  | 23.15\% | 25.32\% | 17.58\% | 18.17\% | 0.00\% |
> |              | Nr. Sol. | 695.60         | 111.38  | 61.37   | 51.52    | 51.95   | 209.96  | 186.91  | 165.83  | 64.14  |
> |              | Time (s) | 537.58         | 629.61  | 2.87    | 2.83     | 3.51    | 23.07   | 22.90   | 23.92   | -      |
> | 15×10 | HV       | 0.4599         | 0.3283  | 0.4447  | 0.4582   | 0.4611  | 0.4653  | 0.4768  | 0.4826  | 0.7106 |
> |              | Gap      | 35.29\%        | 53.81\% | 37.42\% | 35.52\%  | 35.12\% | 34.53\% | 32.90\% | 32.09\% | 0.00\% |
> |              | Nr. Sol. | 399.58         | 55.32   | 63.34   | 53.10    | 59.52   | 250.78  | 212.46  | 213.13  | 90.31  |
> |              | Time (s) | 1652.62        | 1034.72 | 5.15    | 5.01     | 6.05    | 43.59   | 43.11   | 45.12   | -      |
> | 20×10 | HV       | 0.3828         | 0.2572  | 0.5325  | 0.5239   | 0.5500  | 0.5436  | 0.5341  | 0.5660  | 0.6485 |
> |              | Gap      | 40.97\%        | 60.34\% | 17.89\% | 19.22\%  | 15.19\% | 16.18\% | 17.65\% | 12.73\% | 0.00\% |
> |              | Nr. Sol. | 185.30         | 45.69   | 66.12   | 52.29    | 55.78   | 256.21  | 168.87  | 181.97  | 79.90  |
> |              | Time (s) | 2758.37        | 1697.82 | 9.36    | 9.18     | 10.33   | 81.66   | 77.99   | 82.74   | -      |

---

> ### Author Response · Authors · 2025-11-25
> **Response to Reviewer XAsV (2/7)**
>
> **W1 continued**: Lastly, we would like to stress that our proposed method is not limited to the specific objectives studied in this work. Existing DRL-based scheduling work has predominantly focused on single-objective problems that optimize only the simple makespan objective. In contrast, our method provides a general multi-objective framework that can handle a variety of objectives. The reward functions and state features we designed can be adapted to other nonincreasing or nondecreasing objectives as well. We already demonstrate this using 5 frequently occurring objectives in the literature that are closely linked to practical relevance. However, only a limited subset of objectives can be studied in a single paper. We therefore greatly encourage future work to explore additional objectives found in the literature and in practice.
>
> **W2**: In Appendix I and Figures 1 and 2 of the original manuscript (Appendix J and Figures 3 and 4 in the revised version), we already provided visualizations of the learned Pareto fronts for several objective combinations, **which have highlighted how the objectives compete with each other**. These plots illustrate that the objectives are conflicting and that our method is able to learn a diverse set of trade-off solutions. In the revised version, we **have added additional insights into how the objectives are related** in Section 3.2.
>
> In short, the makespan, flowtime, and tardiness objectives are correlated but still competing. They generally all benefit from shorter processing times, but they emphasize different aspects and may lead to different decisions and different optimal solutions. The cost objective is more strongly competing with the other three, as higher costs typically co-occur with shorter processing times. With the addition of the new earliness objective in the revised version, we introduce another conflicting objective, since this objective may benefit from "slower" assignments. The fact that these objectives are competing is reflected in the shapes of the Pareto fronts in Figures 3 and 4 and in the observation that the metaheuristic baselines struggle to obtain good trade-offs.
>
> Moreover, because the considered objectives are well known in the single-objective and multi-objective scheduling literature, we refer the reviewer to existing surveys and scheduling research for more in-depth discussions of how these objectives are related and why they are relevant [1,2]. We include these guiding references in Section 3.2 of the revised paper for interested readers.

---

> ### Author Response · Authors · 2025-11-25
> **Response to Reviewer XAsV (3/7)**
>
> **W3**: We have further clarified and highlighted the novelty and contributions of our work in the final paragraph of the introduction section. Our paper proposes several contributions:
>
> 1. **Decomposition-based PPO framework.** We propose a novel decomposition-based PPO framework for multi-objective scheduling problems that is both theoretically grounded and practically applicable.
> 2. **Conditional attention-based network architecture.** We introduce a novel conditional attention-based network architecture that outperforms multiple baselines on FJSP across a wide range of problem instances and objectives.
> 3. **Bound-based reward functions and state features.** We design new bound-based reward functions and state features for multiple prevalent objective functions in scheduling. This mechanism can be extended to any nonincreasing or nondecreasing objective during solution construction.
> 4. **Extensive empirical evaluation.** We experimentally show that our approach outperforms strong metaheuristic baselines on a variety of objective combinations and problem instances. Moreover, our approach is directly transferable to other popular scheduling problems such as JSSP and FFSP.
>
> Alongside the direct transferability to other scheduling problems such as JSSP and FFSP, several components of our method are problem independent and transferable. For instance, our decomposition-based PPO algorithm does not depend on problem-specific structure and can be applied to other combinatorial optimization problems without modification. Moreover, the idea of objective bound-based reward functions and state features can be reused for other nonincreasing or nondecreasing objectives in combinatorial optimization problems. Lastly, although we apply our conditional attention mechanism within the dual attention network, this mechanism can be used to inspire and adjust other neural architectures involving message passing or attention, albeit for scheduling or other combinatorial optimization problems.
>
> **Difference from existing multi-objective scheduling.** As described in our related work section, DRL for multi-objective scheduling DRL is underexplored. There exist only a few works and most offer very little value as they use simple vector-based state spaces, which are far from state-of-the-art, and do not perform true multi-objective optimization that aims to find sets of non-dominated solutions. Su et al. [4] provide the best alternative approach, using a hypernetwork to generate policies for different objective preferences. However, their method is limited to a specific MOFJSP scenario with specific objectives. They do not provide a framework that can be readily applied to multiple objective combinations and they miss many components that we introduce. In our work, we put forward components such as the decomposition-based PPO algorithm, novel bound-based reward formulations, multiple bound-based state features, and a conditional attention network, all of which are not present in [4]. Moreover, as we highlighted in our revised manuscript, our DCAN network consistently outperforms the hypernetwork approach.
>
> **Difference from existing NCO frameworks** In addition, in the related work section we have expanded the explanation of why the existing work in neural multi-objective combinatorial optimization for routing is not applicable to the scheduling problems that we consider.
>
> First, these multi-objective routing works leverage routing-specific single-objective neural networks that differ substantially from those used for scheduling. In vehicle routing, the state is relatively simple, consisting of static coordinates of homogeneous nodes. In contrast, scheduling requires a richer graph structure with heterogeneous nodes and relations, as well as dynamic states with more, time-dependent features. Consequently, these routing networks are far from applicable and cannot be directly transferred.
>
> Second, due to their structural simplicity, routing problems typically employ simple episodic rewards reflecting the total distance of a solution and are trained with REINFORCE. In contrast, state-of-the-art scheduling methods require stepwise rewards with dynamic states, which do not work with REINFORCE and instead rely on actor–critic methods such as PPO.
>
> Third, the objectives considered in these multi-objective routing works are simple coordinate-based distances that do not reflect practically relevant objectives that arise in scheduling. Hence, although they demonstrate the promise of multi-objective NCO, existing multi-objective routing methods cannot be applied to the FJSP problems we study.

---

> ### Author Response · Authors · 2025-11-25
> **Response to Reviewer XAsV (4/7)**
>
> **W4**: We want to stress that our bound-based reward functions actually *stabilize* the reward signal, reduce instability, and improve credit assignment compared to standard step-wise rewards, as opposed to introducing instability. The objective values are mainly determined by 2 aspects, operation-machine assignments and operation sequencing. Our bound-based reward functions drastically reduce the noisiness of the reward signal that arises from operation-machine assignments.
>
> As an example, consider a trivial problem where we have two operations. Operation 1 has machine alternatives with processing times 10 and 20, while operation 2 has alternatives with processing times 30 and 40. Using a standard step-wise reward for makespan, if the agent selects the alternative with processing time 20 for operation 1, the immediate reward would be -20. At the same time, selecting the alternative with processing time 30 for operation 2 would yield an immediate reward of -30. Thus, the reward signal would suggest that the choice of the machine alternative for operation 1 is a better action than that for operation 2. However, actually, the opposite is true, as the choice for operation 2 provides the fastest option among the possible choices for that operation, while the choice for operation 1 is the slowest option. In our bound-based reward, this would be appropriately reflected, as the reward for selecting the slower alternative for operation 1 would be -10 (i.e., 10 - 20), while the reward for selecting the faster alternative for operation 2 would be 0 (i.e., 30 - 30). Thus, our bound-based reward correctly indicates that the choice for operation 2 is more beneficial. The same logic can be verified for other objectives such as flowtime, tardiness, costs, and earliness.
>
> Of course, due to the complex interactions in FJSP, the lower bound can be looser or tighter based on the specific instance. However, this is not an artefact that our reward introduces. In fact, it is an issue that is even greater for traditional reward formulations.
>
> To further emphasize the value of our reward functions, we have now included an ablation study in Section 5.4 and Table 5, which we also added below for your convenience. Here, we empirically show that our bound-based reward functions consistently and considerably outperform standard reward formulations. The hypervolumes using our reward definitions are over 50% higher than those using the standard step-wise rewards across all three objective combinations. This clearly demonstrates the advantage of our proposed reward design.
>
> |  |  | Makespan Costs |  | Tardiness Costs |  | Makespan Flowtime Costs |  |
> |---|---|---|---|---|---|---|---|
> | State | Reward | Greedy | Sample | Greedy | Sample | Greedy | Sample |
> |  |  | 0.5068 | 0.5385 | 0.5591 | 0.5417 | 0.4165 | 0.3869 |
> | ✓ |  | 0.5182 | 0.5383 | 0.5734 | 0.5419 | 0.4475 | 0.3908 |
> |  | ✓ | 0.8007 | 0.8131 | 0.7948 | 0.8111 | 0.6036 | 0.6184 |
> | ✓ | ✓ | 0.8083 | 0.8200 | 0.8112 | 0.8306 | 0.6142 | 0.6314 |
>
> **W5**: Please note that in our original submission, the value of the conditional attention mechanism is already demonstrated in our main experiments, through the comparison with the WI-DAN network. As mentioned in the previous response to W4, we have now also included an ablation study in Section 5.4 and Table 5 that highlights the advantage of our proposed bound-based state and reward definitions for different objectives. These ablations further confirm the value of our novel conditional attention network design, state features, and reward functions.

---

> ### Author Response · Authors · 2025-11-25
> **Response to Reviewer XAsV (5/7)**
>
> **Q1**: Our work focuses on the multi-objective flexible job shop scheduling problem (FJSP) through NCO. Even in single-objective NCO for scheduling problems, additional constraints such as setup times have not yet been properly addressed by DRL. This is a relevant and challenging research direction that, in our view, should first be explored in the single-objective setting. For this reason, we do not directly target more complex variants of FJSP with additional constraints in this work.
>
> That said, we emphasize there is *no fundamental limitation* that prevents our method from being extended to handle additional constraints. Our framework is designed as a generalizable framework and therefore does not make any additional assumptions compared to the single-objective DRL approaches. Our learning algorithm is general and does not depend on problem-specific structure, so it can be applied to more complex FJSP variants without modification. The state representation can be extended to include features that reflect extra constraints, and the reward functions are tied to the objectives rather than directly to the constraints. Moreover, our conditional attention mechanism is not limited to the current problem setting.
>
> Looking at recent NCO methods for routing problems with constraints (e.g., [5,6,7]), the most common approaches include action masking and penalty terms in the reward function. Both techniques are fully compatible with our methodology. Action masking can be applied directly during both training and inference without changing our method, and penalty terms can be integrated as part of our objectives/rewards. In fact, this could even synergize with our approach: our multi-objective framework could help end users explore and understand the trade-offs between constraint violations and other objective values. This is an interesting direction for future work, especially once initial constrained single-objective FJSP methods have been studied in more detail.

---

> ### Author Response · Authors · 2025-11-25
> **Response to Reviewer XAsV (6/7)**
>
> **Q2**: We have added the earliness objective as an additional objective in our experiments. In Section 4.1, we define the bound-based reward function for earliness, which uses an upper bound rather than a lower bound. In Appendix K and Table 12, we present experimental results for instances that include the earliness objective. These results, also presented below, show that our approach continues to outperform the metaheuristic baselines, highlighting that the framework and reward design generalize to other objectives as well.
>
> |              |          | Metaheuristics |         | Greedy  |          |         | Sample  |         |         |        |
> |--------------|----------|----------------|---------|---------|----------|---------|---------|---------|---------|--------|
> | Size         |          | NSGA-II        | MOEA/D  | Hyper   | WI-DAN   | DCAN    | Hyper   | WI-DAN  | DCAN    | CP-SAT |
> | 10×5  | HV       | 0.5628         | 0.4785  | 0.3871  | 0.3618   | 0.4333  | 0.4306  | 0.4508  | 0.4773  | 0.6531 |
> |              | Gap      | 13.82\%        | 26.74\% | 40.73\% | 44.60/\% | 33.65\% | 34.07\% | 30.97\% | 26.92\% | 0.00\% |
> |              | Nr. Sol. | 1038.92        | 147.41  | 50.71   | 7.23     | 47.49   | 178.60  | 155.77  | 183.66  | 73.54  |
> |              | Time (s) | 255.13         | 267.81  | 0.92    | 0.95     | 1.24    | 5.89    | 5.87    | 6.12    | -      |
> | 20×5  | HV       | 0.3381         | 0.2561  | 0.3545  | 0.3849   | 0.3895  | 0.3785  | 0.4177  | 0.4147  | 0.5068 |
> |              | Gap      | 33.28\%        | 49.47\% | 30.05\% | 24.05\%  | 23.15\% | 25.32\% | 17.58\% | 18.17\% | 0.00\% |
> |              | Nr. Sol. | 695.60         | 111.38  | 61.37   | 51.52    | 51.95   | 209.96  | 186.91  | 165.83  | 64.14  |
> |              | Time (s) | 537.58         | 629.61  | 2.87    | 2.83     | 3.51    | 23.07   | 22.90   | 23.92   | -      |
> | 15×10 | HV       | 0.4599         | 0.3283  | 0.4447  | 0.4582   | 0.4611  | 0.4653  | 0.4768  | 0.4826  | 0.7106 |
> |              | Gap      | 35.29\%        | 53.81\% | 37.42\% | 35.52\%  | 35.12\% | 34.53\% | 32.90\% | 32.09\% | 0.00\% |
> |              | Nr. Sol. | 399.58         | 55.32   | 63.34   | 53.10    | 59.52   | 250.78  | 212.46  | 213.13  | 90.31  |
> |              | Time (s) | 1652.62        | 1034.72 | 5.15    | 5.01     | 6.05    | 43.59   | 43.11   | 45.12   | -      |
> | 20×10 | HV       | 0.3828         | 0.2572  | 0.5325  | 0.5239   | 0.5500  | 0.5436  | 0.5341  | 0.5660  | 0.6485 |
> |              | Gap      | 40.97\%        | 60.34\% | 17.89\% | 19.22\%  | 15.19\% | 16.18\% | 17.65\% | 12.73\% | 0.00\% |
> |              | Nr. Sol. | 185.30         | 45.69   | 66.12   | 52.29    | 55.78   | 256.21  | 168.87  | 181.97  | 79.90  |
> |              | Time (s) | 2758.37        | 1697.82 | 9.36    | 9.18     | 10.33   | 81.66   | 77.99   | 82.74   | -      |
>
> Non-regular objectives do pose an extra challenge for constructive solution methods, which is also reflected in the slightly larger gap to CP-SAT. However, we emphasize that current single-objective NCO methods have primarily focused on the simple makespan objective. Our work already provides a substantial extension to a variety of relevant objectives and also shows promising results for the earliness objective, despite its non-regular nature. We would greatly encourage future work to further improve the performance of NCO methods on non-regular objectives. Potential directions include post-processing the generated solutions or combining constructive and improvement-based methods.
>
> **Q3**: Contrary to routing problems, REINFORCE is rarely used in scheduling works, and state-of-the-art methods such as DAN rely on actor–critic methods such as PPO, as also highlighted in recent review papers [8,9]. This is mainly due to the need for step-wise rewards and dynamic states in scheduling problems, which do not work well with REINFORCE. There is currently no competitive REINFORCE approach for the FJSP. Hence, we follow the state-of-the-art single-objective scheduling works and build our multi-objective method upon PPO.

---

> ### Author Response · Authors · 2025-11-25
> **Response to Reviewer XAsV (7/7)**
>
> **Q4**: See our answer to W4. We re-emphasize that our bound-based reward functions actually *stabilize* the reward signal, reduce instability, and improve credit assignment compared to standard step-wise rewards, rather than introducing instability. This is reflected in our newly added ablation study in Section 5.4 and Table 5. Our reward formulation leads to more consistent reward signals that better match actual action quality. There is, of course, still variance and complex patterns in the rewards due to interactions in FJSP, but this is not an artefact introduced by our reward; it is inherent to reinforcement learning and complex combinatorial optimization problems solved in a constructive manner. In fact, it is an issue that is even greater for traditional reward formulations.
>
> **Q5**: As mentioned in our response to W5, the value of the conditional attention mechanism is already demonstrated in our main experiments, through the comparison with the WI-DAN network. We have now also included an ablation study in Section 5.4 and Table 5 (which we also presented in our response to W4) that highlights the advantages of our method. These ablations demonstrate the impact of the different components of our approach.
>
> [1] Dauzère-Pérès, S., Ding, J., Shen, L., & Tamssaouet, K. (2024). The flexible job shop scheduling problem: A review. European Journal of Operational Research, 314(2), 409-432.
>
> [2] Xie, J., Gao, L., Peng, K., Li, X., & Li, H. (2019). Review on flexible job shop scheduling. IET collaborative intelligent manufacturing, 1(3), 67-77.
>
> [3] Rabiee, M., Zandieh, M., & Ramezani, P. (2012). Bi-objective partial flexible job shop scheduling problem: NSGA-II, NRGA, MOGA and PAES approaches. International Journal of Production Research, 50(24), 7327-7342.
>
> [4] Frutos, M., Olivera, A. C., & Tohmé, F. (2010). A memetic algorithm based on a NSGAII scheme for the flexible job-shop scheduling problem. Annals of Operations Research, 181(1), 745-765.
>
> [5] Bi, J., Ma, Y., Zhou, J., Song, W., Cao, Z., Wu, Y., & Zhang, J. (2024). Learning to handle complex constraints for vehicle routing problems. Advances in Neural Information Processing Systems, 37, 93479-93509.
>
> [6] Li, T., Zou, H., Wu, J., & Wen, Z. (2025). LMask: Learn to Solve Constrained Routing Problems with Lazy Masking. arXiv preprint arXiv:2505.17938.
>
> [7] Luo, F., Wu, Y., Zheng, Z., & Wang, Z. (2025). Rethinking Neural Combinatorial Optimization for Vehicle Routing Problems with Different Constraint Tightness Degrees. arXiv preprint arXiv:2505.24627.
>
> [8] Smit, I. G., Zhou, J., Reijnen, R., Wu, Y., Chen, J., Zhang, C., ... & Nuijten, W. (2025). Graph neural networks for job shop scheduling problems: A survey. Computers & Operations Research, 176, 106914.
>
> [9] Zhang, X., & Zhu, G. Y. (2025). A literature review of reinforcement learning methods applied to job-shop scheduling problems. Computers & Operations Research, 175, 106929.

---

### Official Review · Reviewer_399M · 2025-10-29

**Soundness:** 3
**Presentation:** 3
**Contribution:** 2
**Rating:** 6
**Confidence:** 3

**Summary:**

The paper presents two extensions to an existing neural network architecture for the FJSSP, namely the Dual Attention Network, enabling it to address multi-objective combinatorial optimization problems. The authors aim to approximate the standard Pareto front and identify the corresponding Pareto set.

The first proposed method is a straightforward extension in which the weighted preference vector of the objectives is concatenated with each operation and machine feature. The rest of the network remains unchanged from the single-objective formulation. The second method integrates the preference vector directly into the update process of the operation and machine embeddings during the attention mechanism.

The authors investigate four optimization objectives: makespan, tardiness, flow time, and cost. The two proposed extensions are evaluated against standard metaheuristic algorithms and a mathematical solver across several benchmark datasets. Additionally, the approach is extended to two related problem types, the classical JSSP and the FFSP (discussed in the appendix).

The results demonstrate that both neural network extensions achieve significantly faster computation times compared to metaheuristic algorithms and solver-based methods across all instance types. However, the metaheuristics outperform the proposed approaches on certain, smaller problem instances, while the advantage of the proposed methods becomes particularly evident for larger instances that are computationally challenging for traditional optimization techniques. It was also shown that the second proposed approach, in which the preference is integrated into the attention mechanism, performs better than the approach where the preference vector is trivially inserted.

The authors want to publish code after acceptance.

**Strengths:**

- Well-motivated problem formulation, clearly highlighting that neural network architectures have rarely been applied to multi-objective variants of the Flexible Job Shop Scheduling Problem (FJSSP).

- Introduction of two new methods that improve solution quality, particularly with respect to inference time: The first method, while not highly innovative, forms a solid baseline. The second method shows clear advancements beyond the baseline.

- Comprehensive evaluation setup: Both methods are tested using greedy and sampling-based inference strategies. This allows a thorough assessment under different inference regimes.

- Novel reward function design: Reward is defined based on the change in the theoretical minimum lower bound before and after an action for each metric. This differs from previous work, which typically uses dense or sparse rewards derived directly from the actual schedule rather than theoretical lower bounds.

- Well-structured and informative appendix: Contains mathematical assumptions, details on key performance metrics, description of the critic network architecture (PPO), and a step-by-step reward calculation.

- Transparent discussion of limitations: Authors acknowledge where their methods underperform on small instances. They provide a convincing explanation why their approach performs better on larger instances.

- Demonstrates understanding of the problem characteristics and adds credibility to the evaluation.

**Weaknesses:**

- Although the authors acknowledge that other deep learning solutions typically train separate models for individual preference vectors, no such method is included in the experiments. Including at least one representative baseline (e.g., policy-based or value-based DRL approaches from recent literature) would provide meaningful context and allow readers to better assess the relative performance and contribution of the proposed methods.

- The description of how the preference vector is incorporated into the second method (architecture with dual inputs) remains unclear. A visual schematic, for example a processing pipeline similar to those used in DAN literature, would help illustrate how the model processes and uses preference information.

- While the reward formulation (based on changes in theoretical lower bounds) is novel, the paper does not empirically justify why this reward is advantageous over more traditional dense or sparse scheduling rewards. A small ablation experiment (for example in the Appendix) comparing the proposed reward against a common baseline reward would strengthen the motivation and demonstrate its impact on learning behavior and performance.

**Questions:**

1. Since the study introduces four distinct optimization objectives, why was the method not evaluated on all four objectives simultaneously? The experiments only include comparisons for selected two-objective and three-objective cases. Was this limitation due to the constraints of the metaheuristic algorithms and the solver, or does it reflect a limitation of the proposed model itself?
2. In Table 3, it would be helpful to see results for additional sampling iterations. Given that the proposed method can generate substantially more samples within the same computational time as the metaheuristic approaches, it is possible that further sampling could enable it to outperform the metaheuristics. Could you clarify why this comparison was not included?
3. For the ablation studies, why was only NSGA-II considered and not the newer NSGA-III? The latter represents a more state-of-the-art approach for multi-objective optimization.
4. The focus of current research in multi-objective scheduling is shifting toward real-world applications, where objectives are defined for more practical use cases such as energy consumption and energy cost. Why were such abstract objectives chosen in this study? No citations were provided to justify the selection of the investigated objectives.
5. There already exist methods that apply deep learning to multi-objective scheduling. Why were these methods ignored? At least some ablation studies, for example on small instances, should have been included as a comparison against the proposed method.
6. The IGD+ metric was mentioned and used in the Appendix to compare different approaches. Why was it not included in all the tables? Since it represents an alternative to the hypervolume metric, wouldn’t it be reasonable to include both for completeness?
7. Both methods do not appear to be overly complex, and it seems straightforward to combine them into a single approach—for instance, DCAN with the feature space of WI-CAN. Why was this not investigated?

---

> ### Author Response · Authors · 2025-11-25
> **Response to Reviewer 399M (1/4)**
>
> **We thank the reviewer for the effort in reading our paper and for their positive evaluation. We are glad that the reviewer recognizes many positive aspects of our work, such as the evaluation setup, methodological design, clarity, and strong experimental results. Below, we address the remaining concerns raised by the reviewer.**
>
> **W1**: We agree with the reviewer's concern regarding the need for an additional baseline. To address this, we have revisited the routing literature to determine the appropriate baseline (note, we have now also included an elaborate discussion on this literature in our related works section). In the literature, we found that initial works on multi-objective NCO for vehicle routing use a "many-to-many" paradigm where they train a separate neural network for each preference. This has several limitations; it scales poorly and it does not allow us to adapt preference weights to other values. Later methods, such as hypernetwork [1] and single-model approaches [2,3], have been shown to be better. Our approach is in the single-model category. We have now implemented a hypernetwork-based baseline inspired by [1], which has shown strong performance in multi-objective vehicle routing problems and forms the basis of [4]. This baseline uses a hypernetwork that generates actor policies for different objective preferences, allowing it to handle multiple objectives. It thus provides a relevant "multi-model" baseline, while being more efficient and scalable than training separate models for each preference, and therefore establishes a strong comparative baseline. **We have included this hypernetwork baseline in all our experiments, and the results, presented in the revised manuscript, show that our proposed approach consistently outperforms it across all experimental settings**.
>
> **W2**: Thank you for this suggestion, we agree that a figure alongside the textual definitions can help the readers' understanding of the architecture. To address this, we included Figures 1 and 2 in Appendix D to illustrate the network architectures. These figures outline both the general network designs and the conditional operation and machine message attention blocks, which enhance the readability of this paper.
>
> **W3**: We have included an ablation study on the impact of using a bound-based reward versus simple step-wise reward function, which is common, and on the impact of the state features in the revised manuscript (see Section 5.4 and Table 5). The results demonstrate that both the bound-based reward functions and state features contribute positively to the performance of our approach. In particular, the reward functions have a significant impact, as removing them leads to a notable drop in performance across all objective combinations (for example, HV drops from 0.6314 to 0.3869 for the 3-objective problem). This is sensible, as our approach considerably smooths the reward signal and improves credit assignment compared to using standard stepwise rewards. The ablation study therefore strengthens our claim that these components are important for achieving strong performance in multi-objective scheduling problems.

---

> ### Author Response · Authors · 2025-11-25
> **Response to Reviewer 399M (2/4)**
>
> **Q1**: We followed common practice in neural multi-objective combinatorial optimization works, which typically present experiments with 2 or 3 objectives. We also wanted to highlight a representative subset of objective combinations, and given the computational expense of the baselines, we had to draw a line somewhere. However, there is no fundamental limitation in our approach that prevents handling more than 3 objectives. In fact, our method can, in principle, be applied to any number of objectives without conceptual restrictions. To demonstrate this, we have included an additional 4-objective experiment in the revised manuscript (together with a new earliness objective suggested by another reviewer). These results are shown in the new Appendix K and Table 12 (and also added below for you convenience). Our approach continues to perform well compared to the DRL and metaheuristic baselines, while only incurring a slight increase in runtime (as we use 120 preference vectors for the 4-objective case).
>
> |              |          | Metaheuristics |         | Greedy  |          |         | Sample  |         |         |        |
> |--------------|----------|----------------|---------|---------|----------|---------|---------|---------|---------|--------|
> | Size         |          | NSGA-II        | MOEA/D  | Hyper   | WI-DAN   | DCAN    | Hyper   | WI-DAN  | DCAN    | CP-SAT |
> | 10×5  | HV       | 0.5628         | 0.4785  | 0.3871  | 0.3618   | 0.4333  | 0.4306  | 0.4508  | 0.4773  | 0.6531 |
> |              | Gap      | 13.82\%        | 26.74\% | 40.73\% | 44.60/\% | 33.65\% | 34.07\% | 30.97\% | 26.92\% | 0.00\% |
> |              | Nr. Sol. | 1038.92        | 147.41  | 50.71   | 7.23     | 47.49   | 178.60  | 155.77  | 183.66  | 73.54  |
> |              | Time (s) | 255.13         | 267.81  | 0.92    | 0.95     | 1.24    | 5.89    | 5.87    | 6.12    | -      |
> | 20×5  | HV       | 0.3381         | 0.2561  | 0.3545  | 0.3849   | 0.3895  | 0.3785  | 0.4177  | 0.4147  | 0.5068 |
> |              | Gap      | 33.28\%        | 49.47\% | 30.05\% | 24.05\%  | 23.15\% | 25.32\% | 17.58\% | 18.17\% | 0.00\% |
> |              | Nr. Sol. | 695.60         | 111.38  | 61.37   | 51.52    | 51.95   | 209.96  | 186.91  | 165.83  | 64.14  |
> |              | Time (s) | 537.58         | 629.61  | 2.87    | 2.83     | 3.51    | 23.07   | 22.90   | 23.92   | -      |
> | 15×10 | HV       | 0.4599         | 0.3283  | 0.4447  | 0.4582   | 0.4611  | 0.4653  | 0.4768  | 0.4826  | 0.7106 |
> |              | Gap      | 35.29\%        | 53.81\% | 37.42\% | 35.52\%  | 35.12\% | 34.53\% | 32.90\% | 32.09\% | 0.00\% |
> |              | Nr. Sol. | 399.58         | 55.32   | 63.34   | 53.10    | 59.52   | 250.78  | 212.46  | 213.13  | 90.31  |
> |              | Time (s) | 1652.62        | 1034.72 | 5.15    | 5.01     | 6.05    | 43.59   | 43.11   | 45.12   | -      |
> | 20×10 | HV       | 0.3828         | 0.2572  | 0.5325  | 0.5239   | 0.5500  | 0.5436  | 0.5341  | 0.5660  | 0.6485 |
> |              | Gap      | 40.97\%        | 60.34\% | 17.89\% | 19.22\%  | 15.19\% | 16.18\% | 17.65\% | 12.73\% | 0.00\% |
> |              | Nr. Sol. | 185.30         | 45.69   | 66.12   | 52.29    | 55.78   | 256.21  | 168.87  | 181.97  | 79.90  |
> |              | Time (s) | 2758.37        | 1697.82 | 9.36    | 9.18     | 10.33   | 81.66   | 77.99   | 82.74   | -      |

---

> ### Author Response · Authors · 2025-11-25
> **Response to Reviewer 399M (3/4)**
>
> **Q2**: We have included an analysis of additional sampling for the benchmark instances in Appendix N and Table 15. We also show them below. The results show that increasing the number of samples indeed improves performance at the cost of higher runtimes, but with diminishing returns. This is because, after a certain number of samples, more duplicate solutions are generated. Overall, added sampling can improve performance by several percent, but the improvement is not substantial enough to change the main conclusions of our experiments. Recently, some single-objective NCO works have proposed more advanced sampling techniques to further improve performance. Exploring such techniques in the multi-objective scheduling setting is an interesting direction for future work.
>
> |  |  | Metaheuristics |  | Sample |  |  |  |
> |---|---|---|---|---|---|---|---|
> | Size |  | NSGA-II | MOEA/D | 10 | 100 | 400 | CP-SAT |
> | mk | HV | 0.3416 | 0.2517 | 0.3047 | 0.3258 | 0.3351 | 0.5085 |
> |  | Gap | 32.82\% | 50.49\% | 40.07\% | 35.93\% | 34.09\% | 0.00\% |
> |  | Nr. Sol. | 275.70 | 94.30 | 84.50 | 162.30 | 200.40 | 68.90 |
> |  | Time (s) | 1787.04 | 1122.09 | 44.05 | 411.15 | 1647.12 | - |
> | rdata | HV | 0.6586 | 0.5652 | 0.6367 | 0.6487 | 0.6561 | 0.7296 |
> |  | Gap | 9.73\% | 22.53\% | 12.73\% | 11.10\% | 10.07\% | 0.00\% |
> |  | Nr. Sol. | 8.98 | 6.98 | 9.48 | 11.72 | 14.28 | 15.18 |
> |  | Time (s) | 2120.14 | 1164.22 | 47.79 | 485.12 | 1938.43 | - |
> | edata | HV | 0.6439 | 0.5773 | 0.5689 | 0.5811 | 0.5883 | 0.7137 |
> |  | Gap | 9.79\% | 19.11\% | 20.30\% | 18.58\% | 17.58\% | 0.00\% |
> |  | Nr. Sol. | 10.40 | 6.70 | 7.43 | 9.40 | 10.55 | 16.00 |
> |  | Time (s) | 2106.18 | 1140.89 | 47.78 | 486.01 | 1944.45 | - |
> | vdata | HV | 0.7180 | 0.6133 | 0.7378 | 0.7474 | 0.7524 | 0.7907 |
> |  | Gap | 9.20\% | 22.44\% | 6.69\% | 5.47\% | 4.84\% | 0.00\% |
> |  | Nr. Sol. | 11.70 | 6.50 | 10.90 | 12.60 | 15.15 | 12.03 |
> |  | Time (s) | 2236.94 | 1189.89 | 48.26 | 487.44 | 1957.78 | - |
>
> **Q3**: We followed the most recent multi-objective NCO works [1,2,4], which mainly use NSGA-II and MOEA/D as baselines. Moreover, NSGA-III has not been shown to be universally better than NSGA-II for scheduling problems. Results are often similar, as outlined in works such as [5,6]. Hence, we believe that NSGA-II and MOEA/D provide a representative and strong metaheuristic baseline for our experiments.
>
> **Q4**: Although more traditional multi-objective optimization studies in scheduling focus more and more on objectives such as energy consumption, we emphasize that multi-objective neural combinatorial optimization, especially in scheduling, is still very recent and underdeveloped. As such, we propose methodological advances that can handle a variety of objectives. The objectives we consider offer a representative set that is prevalent in the literature and applicable in many practical scenarios [7,8]. However, our methodology is not limited to these specific objectives. In fact, we strongly encourage future work to explore other objectives. Our approach provides a foundation that demonstrates effectiveness on some of the most popular objectives in the scheduling literature and forms a basis for future works that may explore specific practical scenarios and objective sets.
>
> **Q5**: As described in our related work section, DRL for multi-objective scheduling DRL is underexplored with only a few works. Most offer very little value as they use vector-based state spaces and do not perform true multi-objective optimization that aims to find sets of non-dominated solutions. Su et al. [4] provide the best alternative approach, using a hypernetwork to generate policies for different objective preferences. However, it remains limited to a specific MOFJSP scenario with specific objectives. We agree that this hypernetwork approach is a good and competitive baseline. Therefore, as described in our response to W1, we have implemented the hypernetwork baseline and included it in all our experiments. The results show that our proposed approach consistently outperforms this hypernetwork baseline across all experimental settings.
>
> **Q6**: We did not include the IGD+ metric in the main paper for brevity, as the trends are similar to those observed with the hypervolume. We also did not include the IGD+ values for the results in the appendix, as they did not provide additional insight (similar trend to the hypervolume). However, following your suggestion, we will include these results later when we gather all of them from our cluster that is currently under maintenance.

---

> ### Author Response · Authors · 2025-11-25
> **Response to Reviewer 399M (4/4)**
>
> **Q7**: We also experimented with combining the WI-DAN and DCAN methods. However, this did not lead to improved performance compared to using only the conditional attention mechanism. We believe this is because the conditional attention mechanism already provides a strong way to condition the policy on the objective preferences, making the additional WI mechanism redundant. Hence, we opted to include only the conditional attention mechanism in our final approach for simplicity. For completeness, we have now included the results of the combined method in Appendix O and Table 16. For your convenicence, we also include them below:
>
> |  |  |  | 10x5 |  | 20x5 |  | 15x10 |  | 20x10 |  |
> |---|---|---|---|---|---|---|---|---|---|---|
> |  |  |  | Greedy | Sample | Greedy | Sample | Greedy | Sample | Greedy | Sample |
> | Makespan Costs | DCAN | HV | 0.7104 | 0.7647 | 0.5599 | 0.5724 | 0.7723 | 0.8002 | 0.8083 | 0.82 |
> |  |  | Nr. Sol. | 3.46 | 7.77 | 4.04 | 6.82 | 9.14 | 15.42 | 13.85 | 22.66 |
> |  | WI-DCAN | HV | 0.7255 | 0.7644 | 0.5571 | 0.5716 | 0.7746 | 0.8012 | 0.8122 | 0.8209 |
> |  |  | Nr. Sol. | 4.4500 | 8.4500 | 4.3100 | 6.8600 | 9.6000 | 16.2200 | 15.6100 | 24.7200 |
> | Tardiness Costs | DCAN | HV | 0.746 | 0.8272 | 0.6396 | 0.67 | 0.8094 | 0.8338 | 0.8112 | 0.8306 |
> |  |  | Nr. Sol. | 4.36 | 12.89 | 10.88 | 19.51 | 17.04 | 32.39 | 20.03 | 34.01 |
> |  | WI-DCAN | HV | 0.7668 | 0.8310 | 0.6359 | 0.66729 | 0.7914 | 0.8211 | 0.8147 | 0.8295 |
> |  |  | Nr. Sol. | 6.7000 | 16.2200 | 8.0800 | 16.4800 | 14.2800 | 29.4800 | 20.3700 | 34.7100 |
> | Makespan Flowtime Costs | DCAN | HV | 0.4647 | 0.513 | 0.4318 | 0.4529 | 0.5793 | 0.6025 | 0.6142 | 0.6314 |
> |  |  | Nr. Sol. | 22.64 | 52.84 | 32.96 | 81.77 | 43.96 | 121.23 | 57.83 | 179.74 |
> |  | WI-DCAN | HV | 0.4988 | 0.5484 | 0.4028 | 0.4307 | 0.5883 | 0.6061 | 0.6015 | 0.6170 |
> |  |  | Nr. Sol | 23.6 | 65.77 | 33.52 | 79.74 | 44.56 | 112.75 | 55.03 | 175.13 |
>
> [1] Lin, X., Yang, Z., & Zhang, Q (2022). Pareto Set Learning for Neural Multi-Objective Combinatorial Optimization. In International Conference on Learning Representations.
>
> [2] Chen, J., Cao, Z., Wang, J., Wu, Y., Qin, H., Zhang, Z., & Gong, Y. J. (2025). Rethinking neural multi-objective combinatorial optimization via neat weight embedding. In The Thirteenth International Conference on Learning Representations.
>
> [3] Fan, M., Wu, Y., Cao, Z., Song, W., Sartoretti, G., Liu, H., & Wu, G. (2024). Conditional neural heuristic for multiobjective vehicle routing problems. IEEE Transactions on Neural Networks and Learning Systems, 36(3), 4677-4689.
>
> [4] Su, C., Zhang, C., Wang, C., Cen, W., Chen, G., & Xie, L. (2024). Fast Pareto set approximation for multi-objective flexible job shop scheduling via parallel preference-conditioned graph reinforcement learning. Swarm and Evolutionary Computation, 88, 101605.
>
> [5] Ishibuchi, H., Imada, R., Setoguchi, Y., & Nojima, Y. (2016, July). Performance comparison of NSGA-II and NSGA-III on various many-objective test problems. In 2016 IEEE Congress on Evolutionary Computation (CEC) (pp. 3045-3052). IEEE.
>
> [6] dos Santos, F., Costa, L. A., & Varela, L. (2023, September). Performance comparison of NSGA-II and NSGA-III on bi-objective job shop scheduling problems. In International Conference on Optimization, Learning Algorithms and Applications (pp. 531-543). Cham: Springer Nature Switzerland.
>
> [7] Dauzère-Pérès, S., Ding, J., Shen, L., & Tamssaouet, K. (2024). The flexible job shop scheduling problem: A review. European Journal of Operational Research, 314(2), 409-432.
>
> [8] Xie, J., Gao, L., Peng, K., Li, X., & Li, H. (2019). Review on flexible job shop scheduling. IET collaborative intelligent manufacturing, 1(3), 67-77.

---

> ### Author Response · Authors · 2025-12-03
> **Added IGD+ Values**
>
> Dear Reviewer 399M,
>
> As we promised, we have now added all the IGD+ values for Tables 9-15. Additionally, we have updated Tables 6-8 to include the IGD+ values for the hypernetwork baseline as well. As previously discussed, the IGD+ metrics provide insights consistent with the hypervolume results. Readers can now verify this through the updated tables, which improves the completeness of our paper. Thank you again for this helpful suggestion.

---

### Official Review · Reviewer_oBEa · 2025-10-29

**Soundness:** 2
**Presentation:** 3
**Contribution:** 2
**Rating:** 4
**Confidence:** 3

**Summary:**

This paper proposes a decomposition-based learning method to solve multi-objective flexible job shop scheduling (FJSP), where each subproblem associated with a preference vector. They define two neural architectures based on the DAN architecture from the previous literature for single-objective FSJP. WI-DAN concatenates the preference vector to the feature vector whereas DCAN leverages dual attention with conditional operation message attention block and conditional machine message attention block. The authors evaluate the performance of their proposed method on a variety of FSJP benchmarks, comparing with two commonly used multi-objective evolutionary algorithms, to show effectiveness.

**Strengths:**

1. The paper is easy to read and the proposed method makes sense for multi-objective optimization.

2. Empirical results seem promising for the given method.

**Weaknesses:**

1. There have been many works on learning for multi-objective combinatorial optimization for other COPs (e.g. routing). This paper does not give sufficient discussion of these related works (e.g. the related work section only mentions a few works for multi-objective FJSP, but not for other COPs at all). Similarly, there’s no discussion on (1) whether the learning method used in this paper has been applied to other COPs, and (2) whether techniques from other COPs can apply to FJSP considered in this paper. The authors should rewrite the related works and discussions to better connect to existing literature.

2. In general, I’m concerned about the novelty of this paper. The decomposition based PPO algorithm seems to be a standard RL algorithm for multi-objective learning, and the neural architecture design seems to be a straightforward extension from the DAN architecture in the previous work. Given this, I’m worried that this paper does not meet the bar for a high quality ML conference like ICLR.

**Questions:**

See the weaknesses. And further, based on the applicability of multi-objective learning methods for other COPs, the author should consider comparing with those applicable learning methods. Currently, I feel like the number of baselines the author compared is too few. And further, I think it will strengthen the paper if the authors can try to apply their proposed method to more scheduling variants beyond FJSP.

---

> ### Author Response · Authors · 2025-11-25
> **Response to Reviewer oBEa (1/3)**
>
> **We thank the reviewer for the effort in reading our paper and for recognizing the positive results obtained by our proposed method. We are also glad that the reviewer found the paper easy to read and commented positively on its methodological soundness. Below, we address the raised comments in turn.**
>
> **W1**: Prior work on multi-objective neural combinatorial optimization has focused mainly on routing problems. As highlighted in the fourth paragraph of our introduction, these methods are not directly applicable to the scheduling problems we consider, for several reasons.
>
> First, existing multi-objective routing methods rely on routing-specific single-objective neural networks that differ substantially from those used for scheduling. In other words, the graph structures underlying scheduling problems cannot be effectively represented by the neural network architectures typically used in vehicle routing. In vehicle routing, the state is relatively simple, consisting of static coordinates of homogeneous nodes. In contrast, scheduling requires a richer graph structure with heterogeneous nodes and relations, as well as dynamic states with more, time-dependent features. Consequently, the vehicle routing neural networks are far from applicable and cannot be directly transferred.
>
> Second, due to their structural simplicity, routing problems typically employ simple episodic rewards reflecting the total distance of a solution and are trained with REINFORCE. In contrast, state-of-the-art scheduling methods require stepwise rewards with dynamic states, which do not work with REINFORCE and instead rely on actor–critic methods such as PPO.
>
> Third, the objectives considered in these multi-objective routing works are simple coordinate-based distances that do not reflect practically relevant objectives that arise in scheduling. Hence, although they demonstrate the promise of multi-objective NCO, existing multi-objective routing methods cannot be applied to the FJSP problems we study. For example, the flowtime objective in FJSP must be carefully designed to reflect the true effect of an action at each dispatching step on the eventual flowtime relative to other actions, which is more complex than simple calculation of coordinate-based distances at the end of an episode.
>
> To clarify this more explicitly in the manuscript, **we have added a discussion of these works in a new final paragraph of the related work section**. Here we also go in-depth on which types of approaches they are and why they are not applicable. In addition, we recognize that the hypernetwork approach used for multi-objective routing and knapsack problems by Lin et al. [1] can, in principle, be transferred to scheduling problems without many modifications. Therefore, **we implemented a hypernetwork baseline inspired by [1] and included it in all our experiments in the revised paper (see Section 5.1 and 5.2 with Tables 1,2 and 3 as well as appendices H, I, K, and L with Tables 9-13), showing that our approach consistently and significantly outperforms it**. For example, our approach achieves between 7 and 20 percentage points smaller gap than that of hypernetwork for the 10x5 to 20x10 instances, with similar runtime. For 30x10 and 40x10 instances, this increases to up to 40 percentage points for the 3-objective problem.
>
> **Applicability of our approach to other COPs.** Regarding the applicability of our approach to other combinatorial optimization problems, we first note that the FJSP we address is a generalization of several popular scheduling problems, including the job shop scheduling problem (JSSP) and the flexible flow shop scheduling problem (FFSP). Thus, by solving FJSP, our approach is already directly applicable to these problems without any modifications. In our original paper, we have conducted experiments on JSSP and FFSP, two different COPs from FJSP domain. The results in Appendix H (Appendix G in original submission) and Tables 9 and 10 (8 and 9 in original submission) have shown that our method is highly competitive on small instances and achieves about a 10 percentage point better gap than the metaheuristic baselines for 20x20 JSSP and 20x4 FFSP instances already, with the relative advantage again increasing with the problem size.

---

> ### Author Response · Authors · 2025-11-25
> **Response to Reviewer oBEa (2/3)**
>
> **W1 continued**: Beyond this, several components of our method are problem independent and transferable. For instance, our decomposition-based PPO algorithm does not depend on problem-specific structure and can be applied to other combinatorial optimization problems without modification. Moreover, the idea of objective bound-based reward functions and state features can be reused for other nonincreasing or nondecreasing objectives in combinatorial optimization problems. Lastly, although we apply our conditional attention mechanism within the dual attention network, this mechanism is general and can be incorporated into other neural architectures involving message passing or attention.
>
> So, we have addressed multiple scheduling variants, being the JSSP, FFSP and FJSP using our method. We believe multiple components of our approach are more broadly applicable and can transfer to different optimization problems. Although this is beyond the focus of this paper, in which we mainly focus on scheduling problems, we intend to apply our approach to other relevant optimization problems in the future. We have clearly noted this future work in the conclusion of the revised paper.
>
> **W2**: Our paper presents several contributions that offer clear novelty:
>
> 1. **Decomposition-based PPO framework.** We propose a novel decomposition-based PPO framework for multi-objective scheduling problems that is both theoretically grounded and practically applicable.
> 2. **Conditional attention-based network architecture.** We introduce a novel conditional attention-based network architecture that outperforms multiple baselines on FJSP across a wide range of problem instances and objectives.
> 3. **Bound-based reward functions and state features.** We design new bound-based reward functions and state features for multiple prevalent objective functions in scheduling. This mechanism can be extended to any nonincreasing or nondecreasing objective during solution construction.
> 4. **Extensive empirical evaluation.** We experimentally show that our approach outperforms strong metaheuristic baselines on a variety of objective combinations and problem instances. Moreover, our approach is directly transferable to other popular scheduling problems such as JSSP and FFSP.
>
> All of the above contributions are novel, and to the best of our knowledge, they have not previously appeared in the scheduling literature. We have clarified and emphasized these contributions more explicitly in the introduction section of the revised paper. As explained in our response to W1, these contributions also offer transferable insights and techniques for other combinatorial optimization problems, such as JSSP, FFSP, and FJSP. We also note that some works in top-tier conferences introduce relatively limited extensions, for example, adding only a single additional condition to an existing model [2], or partially modifying the decoder part of an existing network [3]. In contrast, our work provides substantially broader contributions, including advancements in neural network architecture, reward design, and the development of a multi-objective PPO training algorithm. Therefore, we believe the novelty of our approach is substantial and has the potential to be recognized by the research community.
>
> **Neural architecture design.** We considerably extend the DAN framework by introducing a conditional attention mechanism, new reward and state formulations, and a decomposition-based training algorithm suitable for multi-objective optimization. Whereas DAN is a single-objective approach that handles only the simple makespan objective, our work provides a state-of-the-art framework that can handle a variety of practically and scientifically relevant objectives for various scheduling problems, across different instance sizes and objective combinations, and achieves strong empirical performance compared to the baselines. Extensive results have shown that we outperform baseline metaheuristics by up to 20 percentage points gap difference on regular instance and up to 40 percentage points for large instances, while also being considerably faster. In addition, we consistently beat the hypernetwork baseline and maintain performance across different instance sets as well as the JSSP and FJSP.

---

> ### Author Response · Authors · 2025-11-25
> **Response to Reviewer oBEa (3/3)**
>
> **W2 continued**: The fact that our approach builds upon a strong single-objective method as a base does not undermine its novelty or our contributions. On the contrary, building on strong single-objective methods is standard practice in multi-objective optimization. For example, metaheuristic methods such as NSGA-II and MOEA/D build upon strong single-objective evolutionary algorithms. Similarly, recent NCO works in routing published at ICLR and related conferences, including multi-objective routing methods [1,2], also build upon strong single-objective NCO methods. Our approach follows this established practice and addresses *multiple* aspects, including architecture, training algorithm, state and reward design, to obtain a high-performing multi-objective scheduling method.
>
> Furthermore, we would also like to emphasize that developing each component of our approach is far from trivial. For example, unlike the common practice of using a single episodic reward for an entire instance, we design step-wise rewards for widely used objectives. This design is non-trivial and can meaningfully inspire future research. All of these aspects are novel and have not been systematically developed in reinforcement learning for general scheduling problems.
>
> **Q1**: As mentioned in our response to W1, we identified that the hypernetwork approach used in [1,4], which achieves highly competitive performance in vehicle routing, is applicable to our scheduling problem with only minor modifications. Therefore, we implemented this hypernetwork baseline and included it in all our experiments. The results, presented in the revised manuscript, show that our proposed approach consistently outperforms this hypernetwork baseline across all experimental settings. For example, our approach achieves between 7 and 20 percentage points smaller gap than that of hypernetwork for the 10x5 to 20x10 instances, with similar runtime. For 30x10 and 40x10 instances, this increases to up to 40 percentage points for the 3-objective problem. Please refer to Tables 1 and 2 for detailed results.
>
> The reviewer also asked us to apply our method to other scheduling variants. We would like to stress that we already included experiments on both the JSSP and FFSP in our original submission (see Appendix G and Tables 8 and 9 in original submission and H, 9, and 10 in the revised version). These experiments demonstrate that our approach transfers directly to these popular scheduling problems without any modifications and achieves strong performance. Our approach outperforms all baselines in almost all cases in the benchmark datasets, as we already more elaborately discussed in our answer to W1.
>
> [1] Lin, X., Yang, Z., & Zhang, Q (2022). Pareto Set Learning for Neural Multi-Objective Combinatorial Optimization. In International Conference on Learning Representations.
>
> [2] Chen, J., Cao, Z., Wang, J., Wu, Y., Qin, H., Zhang, Z., & Gong, Y. J. (2025). Rethinking neural multi-objective combinatorial optimization via neat weight embedding. In The Thirteenth International Conference on Learning Representations.
>
> [3] Huang, Z., Zhou, J., Cao, Z., & Xu, Y. (2025). Rethinking Light Decoder-based Solvers for Vehicle Routing Problems. In The Thirteenth International Conference on Learning Representations.
>
> [4] Su, C., Zhang, C., Wang, C., Cen, W., Chen, G., & Xie, L. (2024). Fast Pareto set approximation for multi-objective flexible job shop scheduling via parallel preference-conditioned graph reinforcement learning. Swarm and Evolutionary Computation, 88, 101605.

---

### Official Review · Reviewer_ck3N · 2025-11-01

**Soundness:** 3
**Presentation:** 3
**Contribution:** 3
**Rating:** 8
**Confidence:** 4

**Summary:**

The paper proposes to use DRL for the multi objective FJSSP. It compares two different neural architectures and also compares to standard MOCO algorithms such as NSGAII. For large instances the algorithm that samples using the trained neural networks has good results. The conditional attention network outperforms the preference vector input network.

**Strengths:**

Good experimental results for large instances
Comparison of two neural architectures
Design of the DCAN architecture
Better results than meta-heuristic approaches

**Weaknesses:**

Simple sampling strategy
NSGA-II has better result on public dataset instances for the 3-objective problem

**Questions:**

How does the results of best network evolve with samples?

---

> ### Author Response · Authors · 2025-11-25
> **Response to Reviewer ck3N**
>
> **We thank the reviewer for reading our paper and for their positive evaluation. We are glad that the reviewer recognizes the value of our newly proposed method and its good experimental results compared to the baselines.**
>
> **W1**: Thanks for the comment. It is correct that NSGA-II outperforms our approach on some benchmark instance sets in terms of hypervolume for the 3-objective problem in Table 3. However, as discussed in the paper, the metaheuristic methods are allocated a substantially longer runtime (on the order of thousands of seconds, compared to only dozens of seconds for our approach).
>
> To make the comparison more fair, we also compared our approach to NSGA-II under shorter runtime budgets (although NSGA-II still runs for twice as long as our sampling inference). As shown in Table 14 (Table 12 in original manuscript), our learned policies remain highly competitive, achieving the best performance in 11 out of 12 cases across different objectives and problem sizes.
>
> Moreover, as shown in Table 8 (Table 7 in the original manuscript), our approach actually outperforms NSGA-II even under the longer runtime in terms of the IGD+ value, another important metric, which further demonstrates the competitiveness of our approach. All  these results indicate that our approach generally outperforms NSGA-II in terms of different metrics.
>
> **Q1**: We refer to Table 4, which illustrates how the number of preferences impacts the performance of our approach. As can be seen, increasing the number of preferences generally improves performance, but with diminishing returns as the added preferences differ less and less from the preferences that are already present. In addition, we added new experiments showing the effect of using more samples in Appendix N and Table 15, which we also added below. We find that the performance does improve and the DRL approach achieves several percentage points performance improvements when given more sampling iterations. However, like the number of preferences, it does have diminishing returns. This is due to the fact that after a certain number of samples, more duplicate solutions are produced. It would be an interesting topic for future research to develop more advanced sampling techniques in the multi-objective FJSP context that reduce the diminishing returns effect.
>
> |  |  | Metaheuristics |  | Sample |  |  |  |
> |---|---|---|---|---|---|---|---|
> | Size |  | NSGA-II | MOEA/D | 10 | 100 | 400 | CP-SAT |
> | mk | HV | 0.3416 | 0.2517 | 0.3047 | 0.3258 | 0.3351 | 0.5085 |
> |  | Gap | 32.82\% | 50.49\% | 40.07\% | 35.93\% | 34.09\% | 0.00\% |
> |  | Nr. Sol. | 275.70 | 94.30 | 84.50 | 162.30 | 200.40 | 68.90 |
> |  | Time (s) | 1787.04 | 1122.09 | 44.05 | 411.15 | 1647.12 | - |
> | rdata | HV | 0.6586 | 0.5652 | 0.6367 | 0.6487 | 0.6561 | 0.7296 |
> |  | Gap | 9.73\% | 22.53\% | 12.73\% | 11.10\% | 10.07\% | 0.00\% |
> |  | Nr. Sol. | 8.98 | 6.98 | 9.48 | 11.72 | 14.28 | 15.18 |
> |  | Time (s) | 2120.14 | 1164.22 | 47.79 | 485.12 | 1938.43 | - |
> | edata | HV | 0.6439 | 0.5773 | 0.5689 | 0.5811 | 0.5883 | 0.7137 |
> |  | Gap | 9.79\% | 19.11\% | 20.30\% | 18.58\% | 17.58\% | 0.00\% |
> |  | Nr. Sol. | 10.40 | 6.70 | 7.43 | 9.40 | 10.55 | 16.00 |
> |  | Time (s) | 2106.18 | 1140.89 | 47.78 | 486.01 | 1944.45 | - |
> | vdata | HV | 0.7180 | 0.6133 | 0.7378 | 0.7474 | 0.7524 | 0.7907 |
> |  | Gap | 9.20\% | 22.44\% | 6.69\% | 5.47\% | 4.84\% | 0.00\% |
> |  | Nr. Sol. | 11.70 | 6.50 | 10.90 | 12.60 | 15.15 | 12.03 |
> |  | Time (s) | 2236.94 | 1189.89 | 48.26 | 487.44 | 1957.78 | - |

---

### Meta-Review · Area_Chair_2iT8 · 2026-01-07

**Summary:**

After carefully checking the paper, the reviews, the rebuttal, and the author-reviewer discussions, I think the weak points outweight the strong points. Two author still concerns the novelty of this work. But I think rebuttal did a good job addressing most of them. Thus, I recommend to accept this paper.

**Reviewer Concerns:**

The score remains unchanged. The authors addressed the issues regarding the ablation studies. I have carefully read the rebuttal.

**Reviewer Scores:**

The score remains unchanged. I have carefully read the rebuttal.

---

### Decision · Program_Chairs · 2026-01-26

Accept (Poster)